



# Identifying Drivers of Surface Ozone Bias in Global Chemical Reanalysis with Explainable Machine Learning

Kazuyuki Miyazaki[1], Yuliya Marchetti[1], James Montgomery[1], Steven Lu[1], and Kevin Bowman[1]

[1]Jet Propulsion Laboratory, California Institute of Technology, Pasadena, CA, USA

**Correspondence:** Kazuyuki Miyazaki (kazuyuki.miyazaki@jpl.nasa.gov)

**Abstract.** This study employs an explainable machine learning (ML) framework to examine the regional dependencies of surface ozone biases and their underlying drivers in global chemical reanalysis. Surface ozone observations from the Tropospheric Ozone Assessment Report (TOAR) network and chemical reanalysis outputs from the multi-model multi-constituent chemical (MOMO-Chem) data assimilation (DA) system for the period 2005–2020 were utilized for ML training. A regression tree-based randomized ensemble ML approach successfully reproduced the spatiotemporal patterns of ozone bias in the chemical reanalysis relative to TOAR observations across North America, Europe, and East Asia. The global distributions of ozone bias predicted by ML revealed systematic patterns influenced by meteorological conditions, geographic features, anthropogenic activities, and biogenic emissions. The primary drivers identified include temperature, surface pressure, carbon monoxide (CO), formaldehyde ($CH_2O$), and nitrogen oxides (NOx) reservoirs such as nitric acid ($HNO_3$) and peroxyacetyl nitrate (PAN). The ML framework provided a detailed quantification of the magnitude and variability of these drivers, delivering bias-corrected ozone estimates suitable for human health and environmental impact assessments. The findings provide valuable insights that can inform advancements in chemical transport modeling, DA, and observational system design, thereby improving surface ozone reanalysis. However, the complex interplay among numerous parameters highlights the need for rigorous validation of identified drivers against established scientific knowledge to attain a comprehensive understanding at the process level. Further advancements in ML interpretability are essential to achieve reliable, actionable outcomes and to lead to an improved reanalysis framework for more effectively mitigating air pollution and its impacts.

## 1 Introduction

Air pollutants such as particulate matter (PM) and ground-level ozone pose a significant risk to human health, ecosystems, and climate. These pollutants are associated with a wide range of adverse health effects, contributing to approximately 8.1 million premature deaths annually in 2021 (Institute, 2024; Fleming et al., 2018). Additionally, ground-level ozone damages vegetation and reduce crop yields (Mills et al., 2018). Accurate assessment and prediction of air pollutant concentrations are essential for



evaluating their environmental impacts and to facilitating the development of effective mitigation strategies (Archibald et al., 2020).

Ground-based monitoring networks, such as the United States Environmental Protection Agency's (EPA) Air Quality System (AQS) and the European Monitoring and Evaluation Programme (EMEP), have provided continuous records of air pollutant concentration. However, these networks are limited in geographic coverage and pollutant types. The data from these ground observation networks, which were compiled under the Tropospheric Ozone Assessment Report (TOAR) activity (Schultz et al., 2017), have been used to study long-term changes in surface ozone. These studies have revealed increases since 2000 in

certain remote and heavily polluted regions of East Asia (Gaudel et al., 2018). Furthermore, the ground observations have been utilized extensively to assess the performance of global atmospheric chemistry models (Young et al., 2018). The second phase of TOAR (TOAR-II) aims to expand the observational network by including additional ground-based stations, especially from new networks in China and India. Despite these advancements, substantial geographic regions, particularly in developing countries where pollution levels are often severe, remain without adequate monitoring. This results in significant gaps in our

understanding of ground-level ozone variability over time and space, limiting our ability to accurately assess and mitigate its impacts.

    Satellite observations, including those from the Ozone Monitoring Instrument (OMI) (Levelt et al., 2018), Infrared Atmospheric Sounding Interferometer (IASI) (Clerbaux et al., 2009), Measurements Of Pollution In The Troposphere (MOPITT) (Deeter et al., 2017a), and the Tropospheric Monitoring Instrument (TROPOMI) (Veefkind et al., 2012), have provided un-

precedented global pictures of air pollutants, including tropospheric ozone (Clerbaux et al., 2009; Bowman, 2013; Miyazaki et al., 2021a) and its precursors (Krotkov et al., 2016; Miyazaki et al., 2017; Bauwens et al., 2020; Elshorbany et al., 2024), over the past few decades. However, these satellite measurements exhibit reduced sensitivity toward the surface, which limits their ability to evaluate global spatial maps of near surface ozone. Recent advancements in satellite products, such as Tropospheric Emissions Spectrometer (TES)-OMI, Atmospheric Infrared Sounder (AIRS)-OMI and IASI-Global Ozone Monitoring

Experiment-2 (GOME-2) multi-spectral retrievals (Fu et al., 2018; Colombi et al., 2021; Okamoto et al., 2023; Pennington et al., 2024), have enhanced the representation of lower tropospheric ozone, particularly in regions with limited ground-based monitoring. Nevertheless, these products still face challenges in accuracy, largely due to the inherent retrieval uncertainties. Their measurements are influenced by various factors such as cloud cover, which can result in spatial gaps and enhanced uncertainties in the data. Furthermore, linking satellite-derived lower tropospheric ozone with surface ozone requires the consid-

eration of intricate chemical and physical processes (Colombi et al., 2021). While satellite measurements of precursor species, such as $NO_2$, VOCs, and CO, provide valuable insights into the chemical regimes and production of ozone (Souri et al., 2024; Elshorbany et al., 2024), they are not directly applicable to the estimation of surface ozone concentrations. Other ground-based measurements, such as ozonesondes, lidar, and aircraft, provide accurate data on free tropospheric and vertical column ozone. These have been used to validate satellite observations. However, they lack the capability to continuously monitor ground-level

ozone.

    Chemical transport models (CTMs) have been employed to generate global or regional maps of atmospheric composition and aerosols, as well as to analyze their evolution. However, CTMs often exhibit substantial biases, such as overestimating



boundary layer ozone by up to 12 ppb in the southeastern United States (Travis et al., 2016; Skipper et al., 2024) and surface
ozone by up to 20 ppb in the southeastern United States and Western Europe (Liu et al., 2022). These biases emerge from
difficulty of simulating complex physical and chemical processes and the inaccuracy of emissions inventories, which are
affected by uncertainties in activity data, emission factors, and spatial-temporal allocations (Janssens-Maenhout et al., 2015).
Identifying the sources of air quality model errors and their underlying mechanisms is vital for improving air quality forecasting
and assessment. However, spatial error patterns often remain unclear due to the limited observational coverage.

Over the past decade, data assimilation (DA) techniques have markedly enhanced our capacity to integrate observational
data, address observational gaps, and provide comprehensive spatiotemporal representations of air pollutant variability at re-
gional to global scales (Lahoz and Schneider, 2014). Previous studies have highlighted the value of simultaneously assimilating
ozone and its precursors to improve surface ozone estimates (Miyazaki et al., 2012, 2019; Sekiya et al., 2024). DA systems
have enabled the long-term integration of multiple satellite observations to generate decadal-scale atmospheric composition
reanalysis products (Inness et al., 2019; Miyazaki et al., 2020a). The global and regional chemical reanalysis products gener-
ated using the state-of-the-art DA systems have been applied in numerous applications, including air quality monitoring and
attribution studies (Lacima et al., 2023; He et al., 2022a; Miyazaki et al., 2014, 2019, 2021b; Sekiya et al., 2023) and human
health impact assessment (Wang et al., 2024). Nevertheless, the quality of chemical DA and reanalysis remains largely limited
by the performance of the underlying model (Inness et al., 2019; Miyazaki et al., 2020c; Sekiya et al., 2024). The potential and
limitations of current chemical reanalysis products have been extensively discussed and summarized by the TOAR-II Chemical
Reanalysis Focus Working Group (Sekiya et al., 2024; Jones et al., 2024; Wang et al., 2024).

In parallel, machine learning (ML) techniques have emerged as powerful tools in the field of Earth sciences (Sun et al.,
2022). ML has been employed to emulate Earth system models, accelerate computational processes, correct physical model
biases, and extend observational datasets. There is growing interest in utilizing ML techniques for air quality assessment and
improving the accuracy of air pollutant predictions (Hickman et al., 2024). For example, ML has been employed to emulate the
GEOS-Chem gas phase chemistry (Keller and Evans, 2019), predict ozone levels during wildfire events Watson et al. (2019),
and generate a high-resolution global distribution of tropospheric ozone from sparse ground-based observations combined with
high-resolution geospatial data (Betancourt et al., 2022). Furthermore, the application of ML techniques has been extended
to the evaluation of nitrogen oxides (NOx) emission inventories (He et al., 2022b), as well as the simulation of simulate
tropospheric oxidant chemistry (Kelp et al., 2022) . Additionally, ML techniques have identified complex relationships among
variables, such as NOx reductions during the period of the global COVID-19 lockdowns (Keller et al., 2021) and the spatial
patterns of meteorological and chemical influences on air quality (Kleinert et al., 2022). Furthermore, ML have been used to
correct physical model biases. For example, gradient-boosted decision trees (e.g., XGBoost) have been utilized to identify and
address potential systematic errors in ozone prediction models (Ivatt and Evans, 2020).

Explainable ML provides an opportunity to uncover the relationships between input variables and model outputs, thereby
offering insights into the drivers of air pollutant and model biases (McGovern et al., 2019). This capability is of particular value
in the context of air quality assessments (Liu et al., 2022), where a comprehensive understanding of the factors contributing
to air pollution and model biases is essential for informed policy-making and the improvement of CTMs. Similarly, ML is





expected to enhance our understanding of bias patterns and the drivers of chemical reanalysis biases, which are often linked to the lack of observational constraints and inherent forecast model errors. The comprehensive information obtained from chemical DA systems provides critical inputs for ML training, thereby enabling improvements in pollution predictions. Furthermore, ML and DA can be effectively combined within a Bayesian framework to enhance physical models and estimate parameters directly from observations (Geer, 2021).

In this study, we develop and apply a novel, explainable ML framework to identify the drivers of ozone bias in decadal chemical reanalysis. By integrating information from chemical reanalysis and ground-based observations, our objective to provide bias-corrected ozone estimates and valuable insights into the factors controlling bias in the reanalysis product. Section 2 outlines the methodology, including the ML framework. Section 3 presents the results, focusing on predicted ozone biases and identified drivers. Section 4 discusses the implications, limitations, and future directions of our approach. Section 5 concludes the study.

## 2 Methodology

### 2.1 Data

#### 2.1.1 Chemical reanalysis MOMO-Chem

This study employs the comprehensive data set on the evolution of atmospheric composition and associated parameters obtained from the MOMO-Chem framework (Miyazaki et al., 2020c). MOMO-Chem assimilated multi-species satellite observations to reproduce three-dimensional atmospheric composition and surface emission distributions. The local ensemble transform Kalman filter (LETKF) (Hunt et al., 2007) was employed, which accounts for errors in the model transport and chemistry at each grid point and time step in the background error covariance. This approach allows for flow-dependent DA analysis and simultaneous optimization of emissions and concentrations, thereby providing comprehensive constraints on the tropospheric chemistry system. Parts of the MOMO-Chem system were utilized in the production of the Tropospheric Chemistry Reanalysis version 1 (TCR-1) (Miyazaki et al., 2015) and version 2 (TCR-2) products (Miyazaki et al., 2020b).

This study utilizes the TCR-2 data set for the period 2005–2020 Miyazaki et al. (2020b) as ML inputs. The TCR-2 data is publicly available and has been used in numerous studies on atmospheric composition and emissions (Kanaya et al., 2019; Miyazaki et al., 2017, 2019, 2021b; Miyazaki and Bowman, 2023). TCR-2 uses Model for Interdisciplinary Research on Climate–Chemical atmospheric general circulation model for study of atmospheric environment and radiative forcing (MIROC-CHASER) (Watanabe et al., 2011) as a forecast model. This model includes tracer transport, wet and dry depositions, and emissions, as well as detailed photochemistry in the troposphere and stratosphere. The model calculates the concentrations of 92 chemical species and 262 chemical reactions (58 photolytic, 183 kinetic, and 21 heterogeneous reactions). TCR-2 has a T106 horizontal resolution (1.125° x 1.125°) with 32 vertical levels from the surface to 4.4 hPa. Meteorological fields used by TCR-2 are nudged towards the 6-hourly ERA-Interim (Dee et al., 2011).



The assimilated data include tropospheric NO2 column retrievals from the QA4ECV version 1.1 level (L2) product for
the Ozone Monitoring Instrument (OMI), GOME-2 and the Scanning Imaging Absorption Spectrometer for Atmospheric
Cartography (SCIAMACHY) (Boersma et al., 2017, 2018). Ozone retrievals are taken from the version 6 level 2 nadir data
obtained from the Tropospheric Emission Spectrometer (TES) (Bowman et al., 2006) and version 4.2 for the Microwave
Limb Sounder (MLS) for pressures of lower than 215 hPa (Livesey et al., 2018). Total column CO data are derived from the
version 7 L2 TIR/NIR product for the Measurements of Pollution in the Troposphere (MOPITT) (Deeter et al., 2017b). It
should be noted that the ozone retrievals assimilated do not contain information on surface ozone. However, the assimilation
of precursors and free-tropospheric and stratospheric ozone provides indirect constraints on surface ozone (Miyazaki et al.,
2019). The performance of TCR-2 has been validated against independent surface and aircraft measurements (Miyazaki et al.,
2020b).

TCR-2 has been evaluated in comparison with other chemical reanalysis products, including Copernicus Atmosphere Mon-
itoring Service (CAMS) (Inness et al., 2019) and GEOS-Chem reanalysis. TCR-2 and CAMS showed reasonable agreement
with each other and with independent observations in the free troposphere and tropospheric column (Huijnen et al., 2020).
The comparison results demonstrate the value of chemical reanalyses for elucidating historical and present-day tropospheric
ozone distributions. However, larger discrepancies have been identified near the surface. A comparison with surface ozone
observations revealed that all reanalyses tend to overestimate surface ozone, with annual mean biases exceeding 15 ppbv in
GEOS-Chem. A seasonal bias analysis indicates that the largest global mean surface ozone bias in GEOS-Chem occurs in
September–November (18.3 ppbv), while the smallest bias is in December–February (14.2 ppbv). The largest mean biases for
TCR-2 and CAMSRA occurred in June-August, at 11.1 ppbv and 6.6 ppbv, respectively, while the smallest mean biases occur
in December–February, at 5.6 ppbv and 2.7 ppbv, respectively (Jones et al., 2024).

In this study, comprehensive information from MOMO-Chem reanalysis outputs, including various meteorological and
chemical variables, was utilized for ML analysis. To enable feasible scientific interpretation, restricting the number of input
parameters used for ML training was a critical step. The selection of input parameters was guided by their relevance to ozone
chemistry and transport, while avoiding redundancy through correlation analysis (see Section 5.2). Following an evaluation
of the sensitivity calculations with varying input parameters, a total of 28 key variables were selected for use in the ML
calculations, as listed in Table 1. Previous studies on the ML application to air pollution (Liu et al., 2022) have emphasized
the importance of basic geographical parameters, such as latitude and day of the year to enhance the predictive performance of
ML models. However, given that the primary objective of this study is to gain insights into model processes and observational
constraints, rather than to optimize prediction accuracy, these basic geographical parameters were excluded from our ML
predictions.

### 2.1.2   TOAR-II ground-based observations

The TOAR-II surface ozone database (Schultz et al., 2017) provides ozone metrics from approximately 23,000 surface sites
globally. The data version used in this study does not encompass the majority of recent datasets from China and India, thereby
constraining the capacity to train the ML model under highly polluted conditions. The ML calculations employed daily max-





**Table 1.** List of ML input parameters derived from MOMO-Chem reanalysis outputs, including key meteorological variables, chemical species, and emissions.

| Variable name | Description | Variable name | Description |
|---|---|---|---|
| BrOX | Bromine oxides | coflux | Carbom monoxide emissions |
| $C_{10}H_{16}$ | Adamantane | NO | Nitric oxide |
| $C_2H_6$ | Ethane | $NO_2$ | Nitrogen dioxide |
| $C_3H_2$ | Propene | noxflux | Nitrogen oxides emissions |
| $C_5H_8$ | Isoprene | olr | Outgoing longwave radiation |
| $CH_2O$ | Formaldehyde | prcp | Precipitation |
| $H_2O_2$ | Hydrogen peroxide | PS | Surface pressure |
| $HNO_3$ | Nitric acid | q | Humidity |
| $HO_2$ | Hydroperoxyl radical | rfluxld | Radiative downward flux long-wave |
| $N_2O_5$ | Dinitrogen pentoxide | rfluxsd | Radiative downward flux short-wave ' |
| $NH_3$ | Ammonia | t | Temperature |
| OH | Hydroxide | u | Zonal wind |
| PAN | Peroxyacetyl nitrate | v | Meridional wind |
| CO | Carbon monoxide | ccover | Cloud cover |

imum 8-hour average (MDA8) ozone concentrations from both urban and non-urban surface sites. However, the reanalysis product, with a spatial resolution of $1.125° \times 1.125°$, is unable to resolve local emissions and chemical processes that drive

ozone variations, particularly in urban areas, as similarly discussed in Young et al. (2018). While the selection of urban sites is of great importance for the evaluation of reanalysis biases, this was not addressed in the current study. Consequently, this limitation may result in the biased estimations of reanalysis performance, particularly in regions where local-scale processes are important.

## 2.2 ML approach

### 2.2.1 Random Forest Model

In order to predict the reanalysis ozone bias with a given set of input variables, we employed a variant of the widely used ensemble tree method, Random Forest (RF) (Breiman, 2001). RF is well-suited for a broad range of modeling and prediction applications due to its robust performance, ease of implementation, and ability to provide explainability metrics for input variables. Specifically, we implemented Quantile Random Forest (QRF) (Meinshausen and Ridgeway, 2006), which modifies

the loss function to predict both the mean and quantile values of the conditional distribution. The quantile values enable the estimation of prediction uncertainties. Furthermore, QRF addresses challenges posed by high-dimensional datasets, mitigating issues related to unstable computations.



### 2.2.2 Explainability metrics

We employed three methods to evaluate explainability: Feature importance (FI), conditional feature contribution (CFC) (Saabas, 2015; Kuz'min et al., 2011), and Permutation importance (PI) (Altmann et al., 2010). As outlined below, the three measures of explainability are complementary and assess distinct aspects of variable importance, including the impact on predicted values, variability, and prediction accuracy. The FI and PI metrics compute the importance of each input variable on a global scale, with respect to each input variable. CFC calculates the importance of each variable at each grid point locally.

FI represents an intrinsic functionality of RF/QRF that quantifies the predictor reduction in variance at each decision tree split based on a specific input variable. These reductions are averaged across all trees in the forest to measures how much variability the true values gain or lose around their mean in a particular leaf/node based on an input variable. The unitless FI values are normalized between 0 and 1, with values closer to 1 indicating greater importance. This metric provides a comprehensive assessment of the global importance of each input variable.

CFC calculates the incremental changes in predicted values at each parent and child tree node of a decision tree for each variable. Subsequently, the values are aggregated over all nodes in a path of a data point, and averaged across all trees in the forest. In contrast to FI, CFC offers a local assessment of importance for each variable at each grid point. This metric can be explored both spatially and temporally, and its units correspond to those of the target variable (e.g., ppb for ozone bias). CFC allows for spatiotemporal exploration of variable importance.

PI is a model-agnostic metric obtained by randomly permuting a single input variable. Following the random shuffling of a variable, the model is refitted and the measure of accuracy is calculated. This is typically done using metrics such as root mean squared error (RMSE), based on a withheld set of data or through cross-validation. The reduction in accuracy when a variable is permuted indicates its importance, independent of other inputs. While PI does not account for cross-correlations between input variables, it can identify independent relationships and highlight inter-variable dependencies.

### 2.2.3 SHapley Additive exPlanations (SHAP)

Additionally, SHapley Additive exPlanations (SHAP) (Lundberg et al., 2020) were employed to attribute the contributions of individual variables to model predictions, which is a state-of-the-art framework for interpreting and explaining ML model outputs. SHAP is rooted in cooperative game theory and distributes the "credit" or influence of each input variable in shaping a model's prediction in an equitable manner. This is achieved by considering all possible permutations of variable combinations and their contributions. SHAP values generalize the concept of CFC, offering a model-agnostic perspective on variable importance. Similar to CFC, SHAP enhances the transparency of model predictions by enabling local attribution of factors influencing each prediction. This facilitates a deeper understanding of the relationships captured by the model and fosters trust in the intricacies of complex ML systems.



## 2.3 Experimental settings

The ML inputs included surface ozone data, MDA8, from the TOAR data sets, which served at the ground truth, along with
outputs from the MOMO-Chem reanalysis. The TOAR station data were gridded in accordance with the MOMO-Chem reanalysis grid resolution of $1.125° \times 1.125°$. For each reanalysis grid box, the median value of surface ozone was calculated using all TOAR stations within the box. To ensure data quality, TOAR observations below 0 ppb or above 150 ppb were excluded. For other reanalysis variables, daytime averages (8–15 local time) were derived from the 2-hourly reanalysis outputs and used in the ML calculations.

To enhance computational tractability and avoid the influence of seasonality, particularly with regard to explainability metrics, a separate QRF model was trained for each month. The cross-validation strategy involved withholding one year of data for all grid locations for the purpose of testing, while the remaining years were used for training. This approach ensured that the spatial coverage was maintained in the training dataset. The ML calculation utilized data over16 years (2005–2020), with an equal number of cross-validation folds corresponding to the withheld years.

Two primary metrics were used to evaluate the ML performance: RMSE and percent variance explained (PVE). RMSE quantifies the average deviation between the actual and predicted values. PVE computes how much overall variance in the data is explained by the ML model, with values ranging from 0 to 1. PVE values closer to 1 indicate that the model effectively captures the underlying structures and patterns in the data.

### 2.3.1 Emulator runs

The ML framework was first evaluated in emulation mode to reproduce the reanalysis MDA8 fields. By leveraging the true global MDA8 state provided by reanalysis for evaluation, this framework allowed for the assessment and optimization of baseline ML performance. Two emulator runs were conducted. The first used global reanalysis MDA8 data for training ($Emu_{gl}$). The second used dense TOAR ozone sampling over three regions only ($Emu_{toar}$). The TOAR sampled area encompassed North America (20°N–55°N, 125°W–70°W), Europe (35°N–65°N, 10°W–25°E), and East Asia (20°N–50°N, 100°E–145°E).

In $Emu_{gl}$, the ML framework utilized global training data, excluding MDA8 itself, to demonstrate ideal performance under comprehensive data coverage. In $Emu_{toar}$, the training was restricted to regions with dense TOAR coverage, allowing for an assessment of the impact of limited observational data on the ability to represent global ozone distributions.

### 2.3.2 Bias predictions

Subsequently, the ML framework is used to predict the reanalysis ozone bias at each grid point on a daily basis. The predicted
bias is validated against the actual bias (reanalysis minus observations) over the TOAR observation locations. Meanwhile, the prediction provides information on the extended global patterns and the drivers of the ozone bias, including areas with no observations.





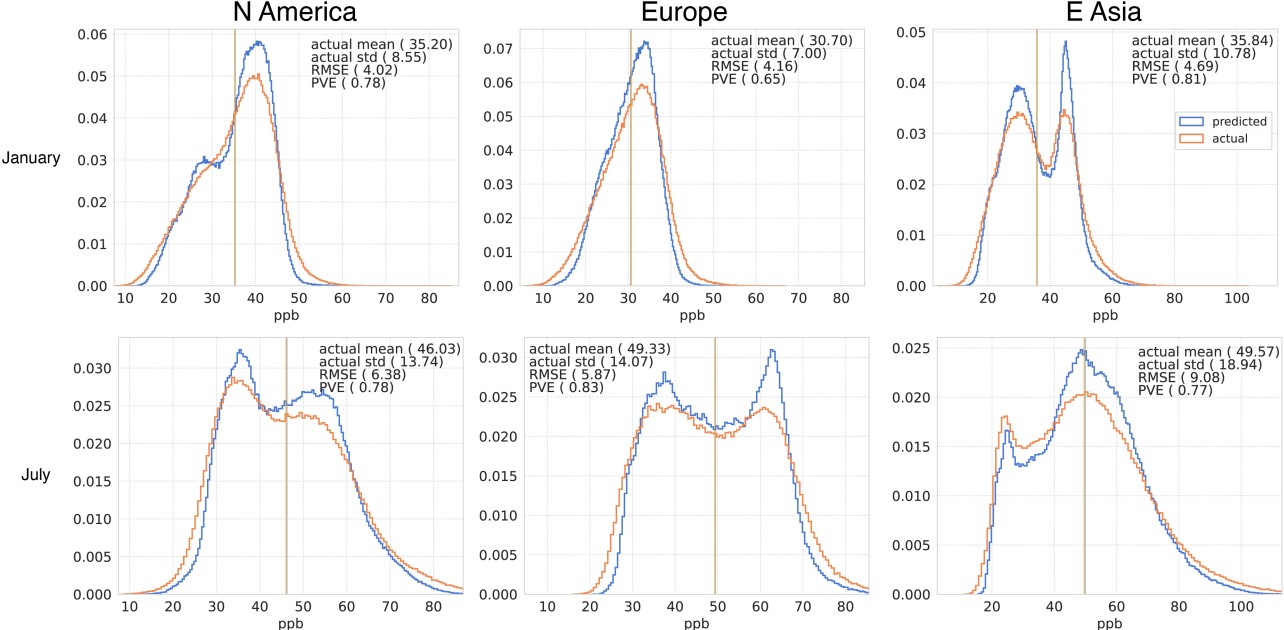

**Figure 1.** Probability distributions of surface ozone for January and July in North America, Europe, and East Asia. Blue lines represent observed ozone concentrations, while red lines represent ML-predicted values. The figure also includes the mean and standard deviation of the observed ozone, as well as the RMSE and PVE of the ML predictions, to evaluate model performance across regions.

## 3  ML performance

### 3.1  Ozone emulator runs

In order to evaluate the overall predictive skill of the ML framework, we first conducted emulator runs using global input data ($Emu_{gl}$). As shown in Fig. 1, the emulator successfully reproduced regional ozone patterns at mid latitudes of the northern hemisphere (NH), with the regional RMSEs ranging from 4.02 to 4.69 in January and from 5.87 to 9.08 ppb in July. The PVE values ranged from 0.65 to 0.83, indicating that the ML model effectively captures the underlying structures and patterns. The global distribution of ozone was also well predicted, with RMSE values below 8 ppb over most land areas and below 5 ppb over oceans at the grid scale (Fig. 2). This confirms the ability of the ML framework to capture the overall spatial variability of ozone. However, notable discrepancies were found in the central Pacific, where relative errors exceeded 30

To examine the influence of limited observational data, an additional emulator run was performed using reanalysis data only from regions with dense TOAR observations (North America, Europe, and East Asia) for ML training ($Emu_{toar}$). In comparison to $Emu_{gl}$, $Emu_{toar}$ demonstrated increased errors in regions such as Central Africa, India, South Asia, Siberia, and the Northwestern Pacific (Fig. 2). This suggests that observational constraints from the TOAR regions, i.e., primarily industrialized areas in the NH mid-latitudes, are inadequate for capturing ozone variability in the tropics and polar regions.





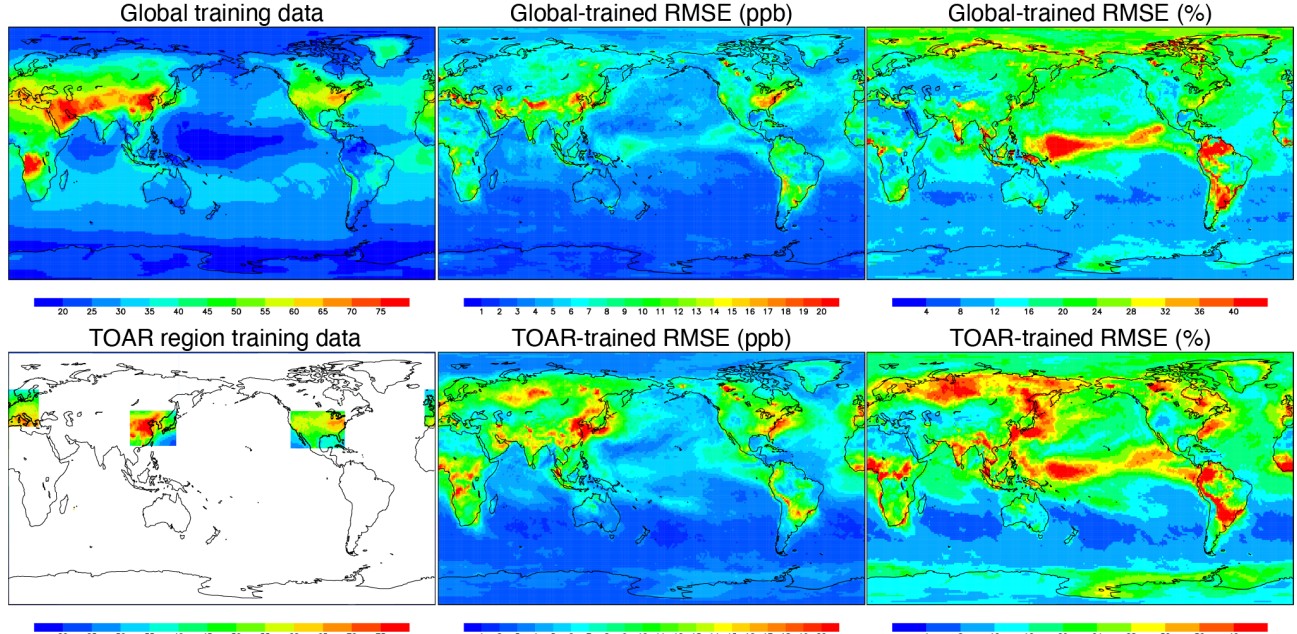

**Figure 2.** Spatial maps of surface ozone in July, derived from (left) MOMO-Chem reanalysis used for training, with RMSE from the emulator run presented in (center) ppb and (right) %. The upper panels depict the results of the ML emulator trained with global MOMO-Chem inputs ($Emu_{gl}$), while the lower panels depict the results of the emulator trained with data limited to TOAR coverage regions ($Emu_{toar}$).

This likely reflects discrepancies in the underlying ozone driving mechanisms. In other regions, the performance of $Emu_{toar}$ was comparable to $Emu_{gl}$, indicating that dense observational coverage in the TOAR regions can inform broader ozone distributions. The comparison between $Emu_{toar}$ and $Emu_{gl}$ provides insights into the robustness and potential uncertainties of ML-predicted biases trained on limited TOAR locations, as discussed further in Section 5.1.

## 3.2 Ozone bias prediction

As depicted in Fig. 3, the actual ozone bias, defined as the reanalysis minus TOAR observations, exhibits a broad Gaussian distribution with mean regional values of 4.93-10.67 ppb in January and 11.3-30.29 ppb in July across the three regions. The bias variability is also greater in July, with standard deviations ranging from 9.19 to 11.54 ppb in January and from 10.31 to 16.46 ppb in July. This reflects the influence of seasonal differences in ozone dynamics. The ML prediction accurately represents he overall actual bias pattern, with RMSE values of 7.8-8.4 ppb in January and 9.6-14.7 ppb in July. Among the regions, East Asia exhibited the largest RMSE values in both seasons. The ML prediction systematically underestimates bias variability across all regions, indicating an underestimation of the occurrence of extreme (both positive and negative) bias values. The larger prediction errors for bias prediction, in comparison to the emulator runs (c.f., Section 3.1), underscore the inherent challenges associated with bias prediction. These challenges likely are likely attributable to errors in the observational data and limitations in the representativeness of the data used for bias estimation.





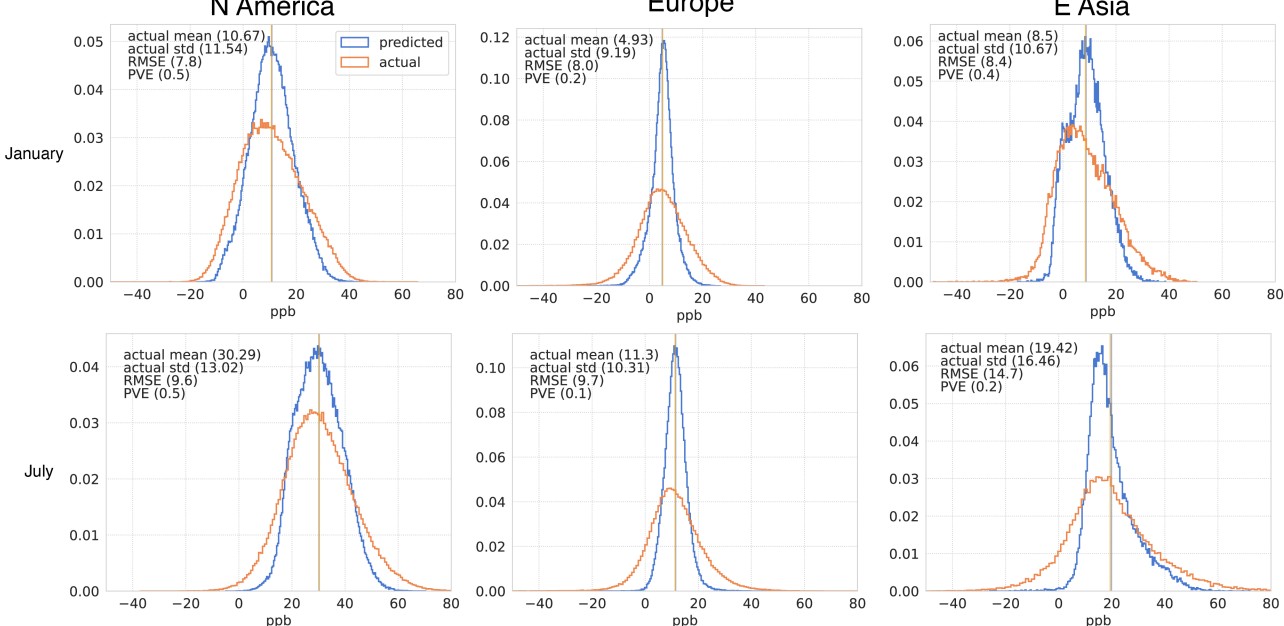

**Figure 3.** Probability distributions of surface ozone bias for January and July in North America, Europe, and East Asia. Blue lines represent actual bias (reanalysis minus TOAR observations), while red lines represent ML-predicted bias values. The figure also includes the mean and standard deviation of the actual bias, as well as the RMSE and PVE of the ML predictions.

As shown in Fig. 4, the reanalysis ozone bias relative to the TOAR observations (i.e., the true bias) exhibits a distinct seasonal pattern, with regional monthly mean positive bias maxima occurring in summer by about 30 ppb for North America in July, 13 ppb over Europe in June, and 24 ppb over East Asia in July. The mean ozone bias is smallest during the winter months across all three regions, with values ranging from approximately 4 to 10 ppb. The smallest bias occurred in January over North America and East Asia, and in February over Europe. The ML predictions well capture the temporal patterns of the actual bias at the regional scale, with temporal correlations of 0.85–0.89. Meanwhile, regional ozone bias also exhibits distinct interannual variability. For example, East Asia experienced larger positive biases during 2005–2008, North America exhibited a slight decreasing trend in biases from 2005 to 2012, and Europe showed greater biases during 2016–2020 compared to earlier years. These variations are likely influenced by a number of factors, including changes in the coverage of ground observations, shifts in the chemical regimes, and discontinuities in the assimilated satellite measurements that were used in the chemical reanalysis (Miyazaki et al., 2020b).

Despite the overall agreement, the ML predictions failed to capture certain anomalies. For example, the ML model overestimates the small bias during the winter of 2010 and the large bias during the summer of 2016 in Europe, while underestimating the large biases during the summers of 2005, 2006, and 2008 in East Asia. These discrepancies may be indicative of an insufficient representation of specific regional processes or limitations in the input data used for ML training. Nevertheless, it is unlikely that these limitations will have a significant impact on the interpretation of the drivers behind the mean bias patterns,





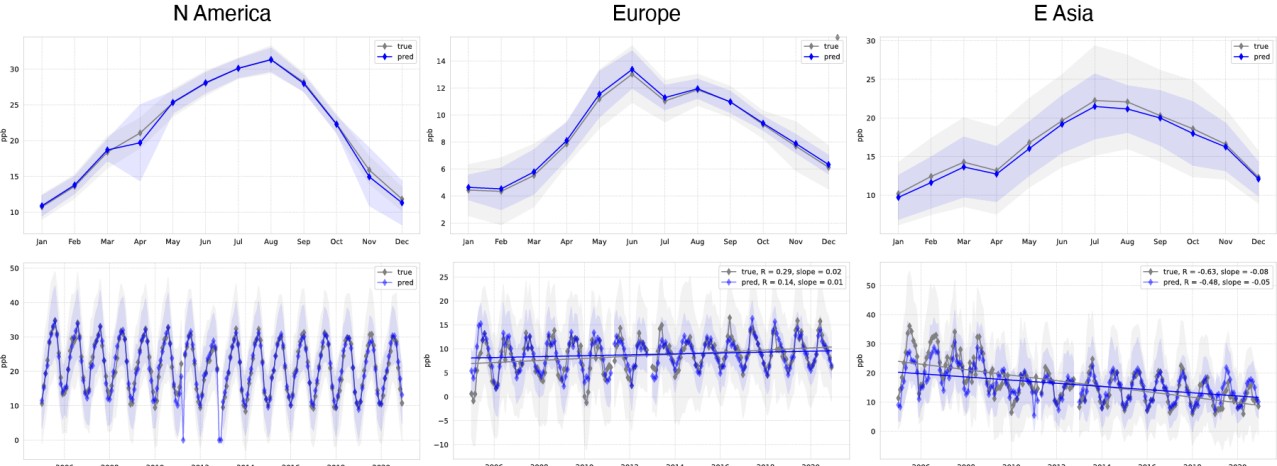

**Figure 4.** (Top) Climatological seasonal variations and (bottom) full time series of actual (black) and ML-predicted (blue) surface ozone bias in ppb over North America, Europe, and East Asia for the period 2005–2020. Shaded areas represent the one-sigma standard deviation for each month, highlighting the variability in the bias.

as the ML framework has demonstrated the capacity to effectively captures the dominant temporal and spatial structures of ozone bias.

### 3.3 The extended global bias patterns

The lack of sufficient global surface observations resulted in a limitation of current knowledge and estimates of surface ozone bias patterns in chemical reanalyses and CTM simulations to specific regions, predominantly in parts of Europe, the United States, and East Asia, as shown in the Fig. 5 upper panels. A comparison with the TOAR observations revealed significant biases in the chemical reanalysis ozone exceeding 20 ppb in southeastern Australia and Mexico in January and 25 ppb in South Korea and the southeastern United States in July.

The application of the ML model presents a valuable opportunity to extend the global understanding of ozone bias patterns. In January, the ML model indicates the presence of widespread positive biases over land at low and mid-latitudes, with values reaching up to 10 ppb over eastern China, 20 ppb over India, and 8 ppb over Western Europe, as illustrated in the Fig. 5 lower panels. Similarly, substantial positive biases are predicted at approximately 20 ppb over Central Africa and 15 ppb over South America. Conversely, ML predicts negative biases of up to 15 ppb at high latitudes north of 60°N.

In July, the predicted positive biases over land are typically larger than those predicted in January. These include biases of up to 30 ppb over the Eurasian continent, eastern and northern parts of North America, Central and Western Africa, and Southeast Asia. The positive biases are especially pronounced in regions such as the southeastern United States, Central Africa, Eastern China, Malaysia, and Indonesia over land. Conversely, negative biases of approximately 10 ppb are predicted for the high latitudes of the southern hemisphere (SH), similar to the negative biases observed in the NH high latitudes in January.



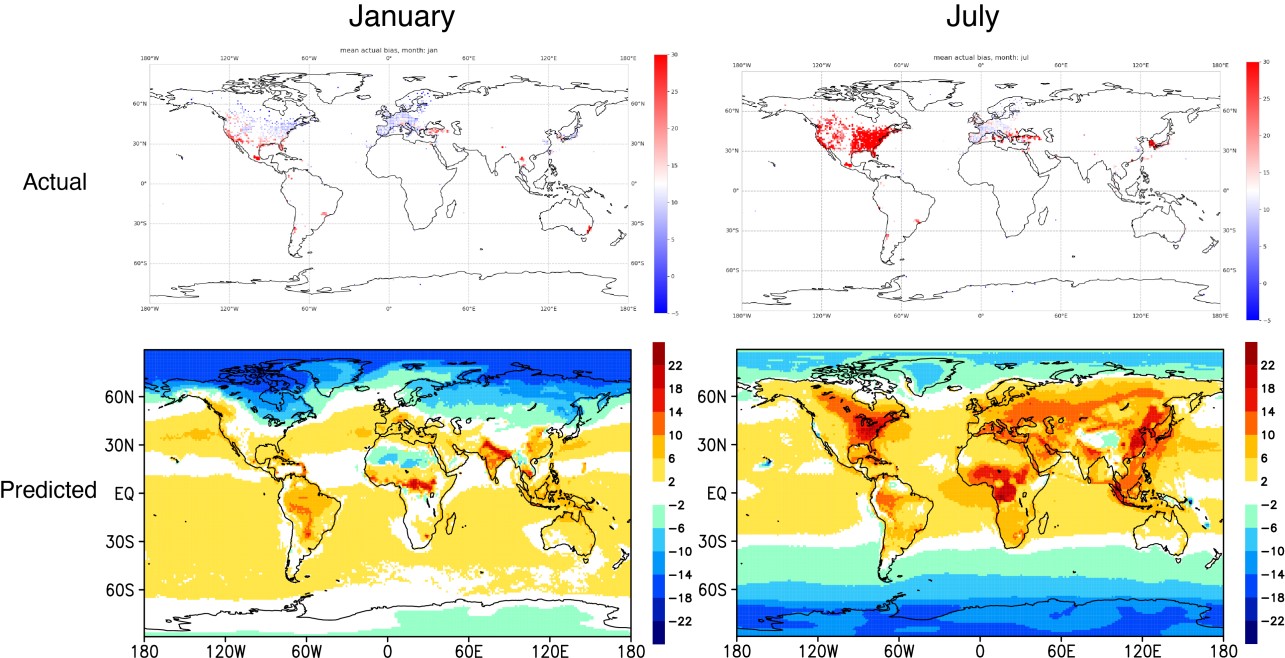

**Figure 5.** Spatial distributions of reanalysis surface ozone bias (in ppb): actual bias at TOAR observation sites (top panels) and ML-predicted bias across the globe (bottom panels) for January (left) and July (right), averaged over the period 2005–2020.

The spatial distribution of the predicted biases appears to correlate with multiple factors, including topography, urbanization, forested areas, and precursor emissions. These factors will be discussed in Section 4. Meanwhile, significant uncertainties are expected in regions where the chemical and physical processes driving ozone biases are not well-represented by ML. This will be discussed in Section 5.1.

## 4 Ozone bias drivers

### 4.1 Regional bias

The explainable ML framework is employed to identify the primary drivers of surface ozone bias. The analysis reveals distinct regional patterns among the top 20 identified drivers on the annual scale (Fig. 6). In most cases, the three approaches yield comparable results with regard to the relative importance assigned to the input variables. Surface pressure emerges as one of the most significant contributors across all three regions, underscoring its capacity to modulate ozone bias through a range of factors, including topographical influences and synoptic-scale weather patterns. Temperature is another critical driver, affecting ozone by influencing chemical reaction rates, local wind patterns, and atmospheric stability. These findings emphasize the fundamental role of meteorological parameters in shaping surface ozone distributions, aligning with previous studies (Weng et al., 2022).



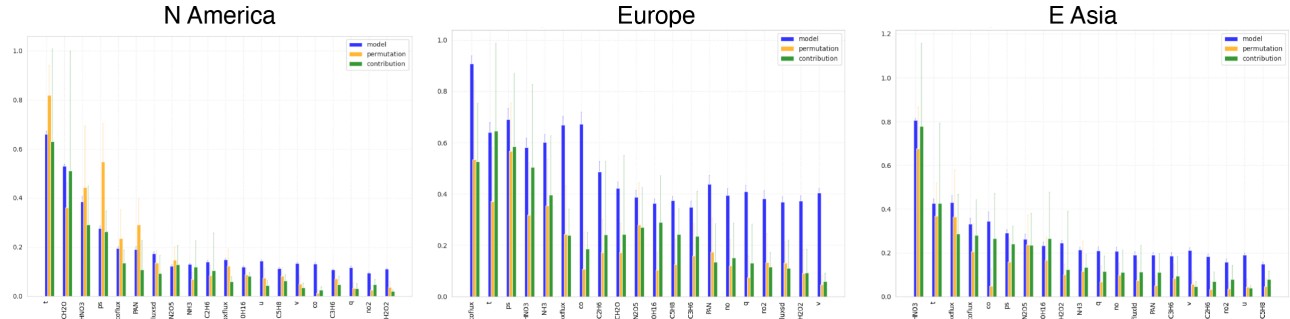

**Figure 6.** Top 20 contributors to regional ozone bias over North America, Europe, and East Asia, identified using three explainability approaches: FI, CFC, and PI.

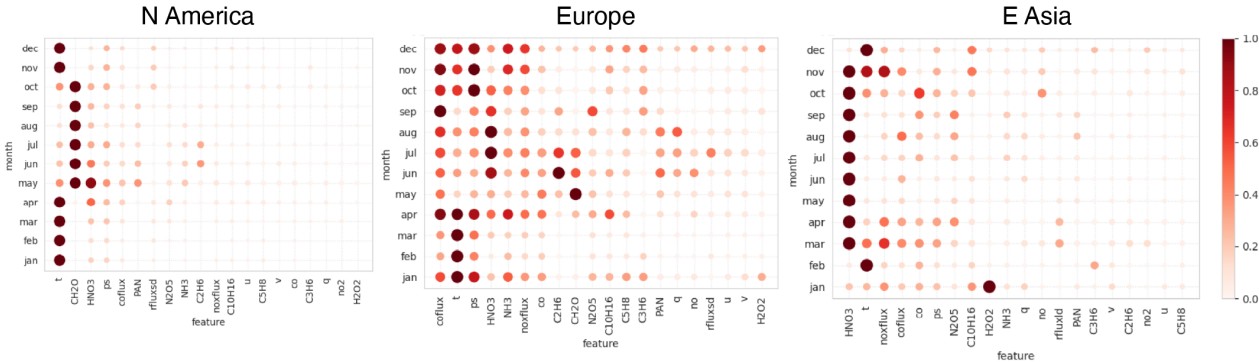

**Figure 7.** Monthly changes in the top contributors to regional ozone bias for North America, Europe, and East Asia, estimated from the combination of the FI and CFC approaches. Bubble size and color represent the magnitude of the impact of each contributor.

310    Other significant contributors include $HNO_3$, NOx emissions, CO emissions, $N_2O_5$, $CH_2O$, and PAN, though their relative importance varies significantly among regions. For instance, East Asia demonstrates more pronounced influences from $HNO_3$, NOx emissions, and CO emissions, which many be attributed to the elevated levels of industrial activity. In contrast, $CH_2O$ exerts the most significant influence in North America, likely reflecting the strong biogenic emissions. PAN, as a reservoir species, also plays a notable role across all regions due to its involvement in ozone formation. These contributors are linked to
315    both anthropogenic and natural processes, including industrial activities, biomass burning, agricultural practices, and wildfires.

As illustrated in Fig. 7, the seasonal variation in ozone bias drivers exhibits pronounced regional characteristics across three regions. As detailed below, these findings highlight the significant regional dependence of seasonal bias drivers, reflecting the complex interplay of meteorological, chemical, and emission-related factors specific to each region. Moreover, common seasonal patterns are evident across regions, such as the influence of temperature during winter and $HNO_3$ during summer,
320    emphasizing the existence of universal processes that govern ozone bias dynamics.





In Asia, $HNO_3$ emerges as a dominant contributor from March to November. The ozone bias is largely influenced by temperature and NOx emissions from October to March, while contributions from $N_2O_5$ peak in summer, $C_{10}H_{16}$ in winter, and $H_2O_2$ in January. Additionally, CO emissions and concentrations exhibit broadly enhanced contributions during the spring and summer months. In Europe, surface pressure and temperature are the primary contributors from October to January. CO emissions show a robust influence throughout the year, with the exception of February and March. Enhanced contributions from $C_2H_6$ and $CH_2O$ are found during early summer months, with $HNO_3$ exerting its largest influence during the summer season. The contributions of $NH_3$, NOx emissions, and CO are moderate throughout the year. In North America, temperature plays a prominent role from November through April, while $CH_2O$ becomes the dominant contributor from May through October. Other notable contributors include $HNO_3$ from late spring through autumn, surface pressure in early summer and winter, and PAN in early summer.

## 4.2 Spatial pattern

This section examines the spatial patterns of ozone bias drivers, classified into primary categories such as meteorological parameters, combustion processes, biogenic and agricultural sources, and reservoir species. By analyzing these spatial distributions, our objective is to identify the predominant contributors to bias in different regions and their associated processes. Spatial maps of selected key contributors are presented in Fig. 8.

### 4.2.1 Meteorological parameters

During the boreal winter months, the contribution of surface pressure is particularly pronounced in northwestern China (Fig. 8a), indicating that the winter Siberian High and the East Asian monsoon circulation exert a significant influence on ozone transport in the region. During the boreal summer months (figure not shown), the area of strong surface pressure contribution shifts southward and is largely diminished over eastern and southern China. This pattern is likely driven by the summer Asian monsoon system, which has been identified as a key factor in surface ozone variability (Li et al., 2018). The sign of the surface pressure contribution reverses between winter and summer in China, with an increasing positive bias in winter and a decreasing positive bias in summer, which partially offsets the positive biases induced by other factors in summer. In contrast, in eastern and southern China, where air pollution is severe, the contribution of surface pressure is much smaller throughout the year.

In Europe, surface pressure plays a significant role in the formation of ozone bias in limited areas, including Spain, northern Italy, and Norway, during winter (Fig. 8a). In these areas, it tends to increase the positive ozone bias. This indicates that surface pressure is associated with local biases, which are influenced by wintertime synoptic weather patterns. During summer (figure not shown), the impact of surface pressure in these regions is reversed, leading to a reduction in the positive bias. However, when compared to other variables, the overall contribution of surface pressure is minimal across Europe on the regional scale. This is reflective of the dominant role of chemical parameters in ozone bias in major polluted areas, similar to the results obtained for southeastern China.

Over the western United States, the contribution of surface pressure displays a complex pattern that follows topographic features. During winter, the surface pressure's contribution tends to increase the positive bias, particularly over the western







**Figure 8.** Spatial maps of the contributions of key parameters to monthly ozone bias, showcasing prominent drivers during specific months. The maps illustrate the influences from meteorological processes, combustion sources, biogenic and agricultural emissions, and NOx reservoir species.

coastal mountainous regions and the northwestern United States (Fig. 8a). During the boreal summer months, this contribution

undergoes a shift, resulting in a reduction of the positive bias across the western half of North America. Additionally, there is a notable influence of surface pressure over the coastal regions of Mexico, the northwestern United States, and the west coast of South America (figure not shown). Among the various parameters, surface pressure has the greatest impact on increasing the positive bias on a regional scale in North America during summer (Fig. 9). This highlights its significant role in shaping ozone bias patterns in specific regions, particularly under the influence of complex topography.

The influence of temperature on ozone bias is driven by a variety of mechanisms, including its impact on gas-phase reaction rates, atmospheric stability, and vertical mixing. The impact of temperature on ozone bias varies by season and latitude. In most cases, positive ozone bias increases at low and mid-latitudes, while at high latitudes, it is reduced (Fig. 8b). The increased





positive bias is particularly pronounced in regions such as the western United States, the Middle East, eastern Africa, the Sahara Desert, and western Australia. The SHAP analysis indicates that temperature is a primary factor contributing to positive bias

over North America (Fig. 9). Furthermore, temperature is identified as the predominant driver of ozone bias at low latitudes in regions such as North Africa, South Africa, the Middle East, eastern South America, western North America, and parts of Siberia during boreal summer (Fig. 10). At high latitudes, temperature plays a dominant role during boreal winter.

Our analysis further demonstrated that radiation exerts a substantial influence on ozone bias through its impact on photo-chemical reactions, thermal balance, and subsequently atmospheric circulation (figure not shown). For instance, photochemical

reactivity at the surface is influenced by incoming solar radiation, which is modulated by humidity, water vapor, and ozone above the surface. Furthermore, ozone levels above the surface impact ozone bias not only through downward transport but also by affecting incoming radiation. The spatial analysis demonstrates that downward short-wave radiative flux at the surface exerts a widespread influence, contributing to increased positive ozone bias at low and mid-latitudes. This effect is especially pronounced over Northern and Central Africa, the Southwestern United States, and South Asia, particularly during the spring

and summer seasons. This highlights the interconnected dynamics of radiative and photochemical processes.

### 4.2.2 Combustion sources

Combustion processes, including industrial activities and wildfires, release CO along with a multitude of other chemical compounds. CO is a primary precursor to ozone and plays a substantial role in chemical ozone production. For example, it has been estimated that ozone produced by wildfires contributes approximately 3.5% of the global total tropospheric ozone pro-

duction (Jaffe and Wigder, 2012). According to the ML analysis, the impact of CO emissions on ozone bias is widespread across extensive emission regions, including East and South Asia, Central Africa, North America, and Europe (Fig. 8d). This indicates that CO emissions exert a considerable influence on ozone bias over and downwind of regions where combustion occurs. Conversely, CO concentrations tend to reduce the positive ozone bias over South America, Central Africa, and Southeast Asia, particularly in areas and periods of active biomass burning (Fig. 8c). This indicates that the effect of extremely high CO

concentrations from wildfires and anthropogenic activities on ozone bias differ from those associated with moderate CO levels. The differing roles of CO emissions and concentrations in ozone bias are not fully understood. Nevertheless, it is likely that the non-linear relationships inherent in chemical processes play a significant role. For example, elevated CO levels may saturate specific chemical pathways or disrupt the balance between ozone production and loss. This can result in divergent impacts depending on atmospheric conditions. These findings highlight the complexity of CO's role in ozone bias.

The production of ozone in urban areas is primarily regulated by chemical regimes that are determined by the concentrations of NOx and volatile organic compounds (VOCs) (Sillman, 1999). However, our ML assessment indicated that the direct impact of NOx emissions on ozone bias was limited (figure not shown). In contrast, NOx reservoir species, such as peroxyacetyl nitrate (PAN) and nitric acid ($HNO_3$) were shown to have significant impacts on ozone bias, as discussed in Section 4.2.5.

Ethane ($C_2H_6$), a hydrocarbon that contributes to ozone formation, has a substantial impact on ozone bias across a range

of geographical regions, including industrial zones, biomass burning areas, and oil basins. Significant impacts were observed over central Africa, northern India, northeastern China, Indonesia, and the northern parts of North America (Fig. 8e). The



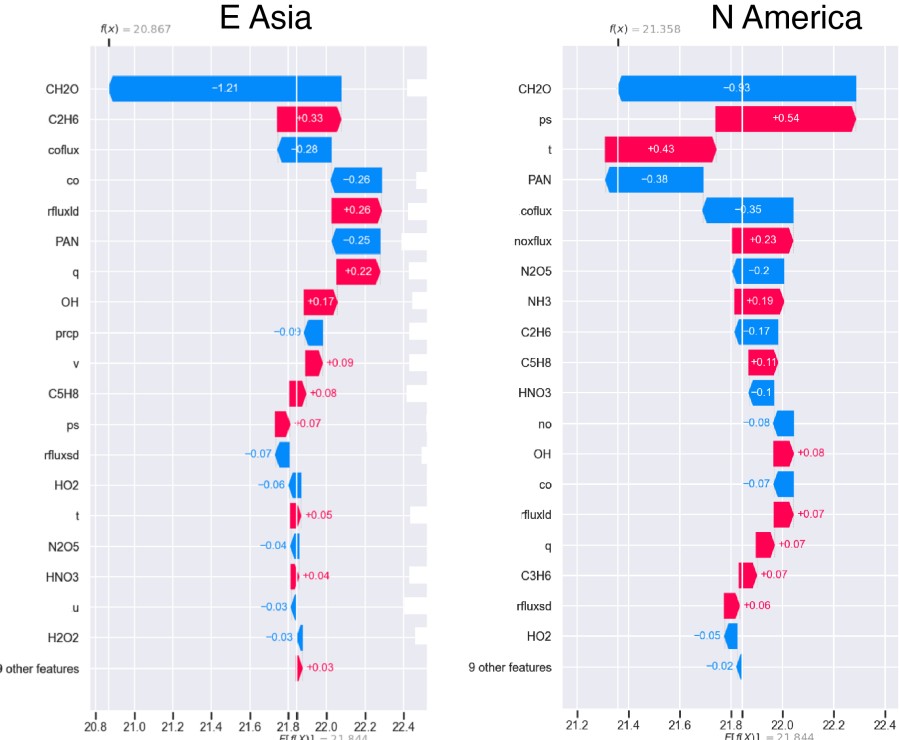

**Figure 9.** SHAP waterfall plots depicting individual parameter contributions to predicted ozone bias in July during 2005–2020. Positive contributions (red) and negative contributions (blue) represent the extent to which each parameter increases or decreases the predicted ozone bias, offering insights into the key drivers of ozone bias variability.

impact of $C_2H_6$ on ozone bias is especially pronounced over the mid-latitudes of the NH during the summer months. In eastern China, $C_2H_6$ notably increases the positive ozone bias, contributing a notable portion of the total bias. According to emission inventories, the global $C_2H_6$ source is estimated to be 13 Tg yr$^{-1}$, with contributions of 8.0 Tg yr$^{-1}$ from fossil fuel production,

2.6 Tg yr$^{-1}$ from biofuel combustion, and 2.4 Tg yr$^{-1}$ from biomass burning (Xiao et al., 2008). However, $C_2H_6$ emissions remain highly uncertain, which could potentially lead to biased ozone estimates. The incorporation of new satellite retrievals of $C_2H_6$ from CrIS (Brewer et al., 2024) into reanalysis frameworks has the potential to reduce uncertainties in $C_2H_6$ emissions and, consequently, improve ozone estimates.

Wildfires emit substantial amounts of chemical compounds, including black carbon, CO, PAN, NOx, and VOCs (Permar

et al., 2021). These emissions impact regional ozone distributions (Cooper et al., 2024; Jin et al., 2023). The elevated contributions of these species in regions with biomass burning regions are evident in the ML calculations. For example, PAN exerts significant impacts in central Africa and South America (Fig. 8i). Formaldehyde ($CH_2O$) also exhibits pronounced seasonal variations driven by biomass burning emissions in tropical regions (De Smedt et al., 2008), exerting a considerable influence on ozone bias over tropical South America, central Africa, and Southeast Asia (Fig. 8f). Furthermore, the presence of





VOCs in wildfire plumes, when combined with the NOx content of urban air, result in a deterioration of urban air quality (Xu et al., 2021). Optimizing wildfire emissions within the reanalysis framework by assimilating supplementary datasets, such as TROPOMI and CrIS CH$_2$O and CrIS PAN data, could facilitate more comprehensive corrections to ozone production associated with wildfire events. While this study focuses on the climatological patterns of ozone bias drivers, future research should assess the impact of individual wildfire events on ozone and its model bias using explainable ML. Such investigations will be

essential for enhancing the accuracy and utility of chemical reanalysis products in capturing event-specific ozone dynamics and their contributions to long-term atmospheric changes.

### 4.2.3 Biogenic sources

Various chemical species are emitted by vegetation, but the relative importance of each biogenic species on ozone remains largely uncertain. This is due to the fact that their contributions are influenced by a range of factors, including meteorological

and chemical conditions, as well as vegetation types. Among these species, isoprene (C$_5$H$_8$) is recognized as one of the most significant VOCs at regional scales due to its strong impact on ozone formation. The contributions of C$_5$H$_8$ exhibit distinct spatial and temporal patterns, which mirror the spatial distribution of its sources and the ozone chemical regimes (Fig. 8h). C$_5$H$_8$ tends to reduce positive ozone biases. However, whether it reduces ozone bias depends on the background bias conditions, which are influenced by many other contributors (Fig. 9). As anticipated, ML highlights the broad impact of C$_5$H$_8$ over land,

notably in forested zones such as central Africa, South Asia, South America, and Australia, where biogenic emissions are pronounced (Guenther et al., 2012). ML uniquely assesses both the sign (positive or negative) and the quantitative contribution of C$_5$H$_8$ to ozone bias, therefore offering deeper insights into its role.

The strong seasonal variations in CH$_2$O are largely attributed to the oxidation of biogenic VOCs. Its impact on ozone bias is particularly pronounced in the eastern United States and southern China during the summer season and in Southeast Asia

during the dry season (Fig. 8f). Consequently, CH$_2$O emerges as a significant contributor to ozone bias in these regions, making it one of the most important bias drivers at regional scales (Fig. 9). In Europe, where biogenic VOC emissions are lower, the contribution of CH$_2$O is less pronounced.

### 4.2.4 Agricultural sources

Ammonia (NH$_3$) is predominantly emitted from agricultural sources, accounting for over 80% of the global total NH$_3$ emis-

sions. This is largely attributed to the pervasive utilization of nitrogen fertilizers in numerous countries. NH$_3$ reacts with other chemical compounds to form aerosol particles, including PM2.5. Elevated amounts of these particles can have severe environmental and health impacts. The impact of NH$_3$ on ozone is more indirect, occurring primarily through alterations in NOx levels and the oxidative capacity of the atmosphere (Pai et al., 2021). The results of the ML analysis indicates a distinct spatial pattern of NH$_3$ influence on ozone bias, with notable contributions observed in regions with elevated agricultural emissions (Fig. 8g).

These areas include Western Europe, eastern and northern India, East China, and the southern and eastern United States. These results highlight the necessity of incorporating complex chemical interactions into the assessment of ozone bias. Moreover,



they indicate that incorporating $NH_3$ emission estimates (Cao et al., 2022) into the reanalysis framework could enhance the efficacy of ozone reanalysis.

### 4.2.5 NOx reservoirs

While NOx emissions and concentrations have a limited impact on ozone bias broadly, the reservoirs, $HNO_3$ and PAN, exert significant effects. $HNO_3$, primarily produced from anthropogenic NO emissions, emerges as an important driver of ozone bias. $HNO_3$ can modulate ozone production efficiency. The enhanced contributions, particularly over eastern Asia, eastern and northern India, eastern Saudi Arabia, and South Africa (Fig. 8j), highlight the critical role of chemical conversion processes between NOx and $HNO_3$ in accurately predicting surface ozone levels.

Similarly, PAN, another reservoir species derived from NOx, is identified as a significant contributor to ozone bias. In colder conditions, the lifetime of PAN is considerably longer, enabling it to be transported over long distances in the free troposphere, where it plays a critical role in the long-range transport of ozone precursors (Shogrin et al., 2023). At the surface level over polluted regions, the contribution of PAN to ozone bias is more localized to its source regions, particularly industrialized areas and regions affected by wildfires. For instance, increased ozone biases are observed over eastern China and the eastern United

States due to the influence of PAN (Fig. 8i). Additionally, PAN contributes considerably to ozone bias in remote regions, such as the tropical oceans situated downwind of regions with high emissions of pollutants, where it tends to reduce positive ozone biases. These findings underscore the significant role of PAN in influencing surface ozone bias both locally and remotely. Furthermore, they highlight the importance of accurately representing NOx-PAN conversion processes in chemical models to improve ozone analysis.

### 4.2.6 Dominant contributing parameters

The ML analysis demonstrates that the principal parameters responsible for surface ozone bias exhibit unique spatial patterns that vary significantly by season (Fig. 10). These systematic patterns reflect the spatial variability of factors such as meteorological conditions, chemical regimes, and natural and industrial activities. The intricate nature of these distributions highlights the challenges in identifying and addressing ozone biases in a comprehensive manner.

In numerous regions and seasons, $CH_2O$ emerges as the predominant contributor, indicating the prevalence of VOC-limited ozone regimes. This finding highlights the need to evaluate emissions inventories and refine the representation of chemical processes involving $CH_2O$ and other VOCs, with the aim of improving the accuracy of reanalysis ozone. Temperature is also s a critical factor, particularly in high latitude regions in both hemispheres during January and October, as well as in regions such as Northern and Southern Africa and the Middle East during July. In these areas, temperature influences various factors,

including chemical reactivity and land and atmospheric conditions. In regions with distinctive topography, such as the NH mid-latitudes and the Andes Mountains in the SH, surface pressure emerges as a dominant factor. This reflects the complex interplay between topography and atmospheric conditions in shaping ozone bias patterns.

In low-latitude land regions, particularly the Middle East, Africa, and Central America, downward shortwave radiation is identified as the most influential parameter in April. In tropical oceanic regions, PAN dominates ozone bias in July, reflecting





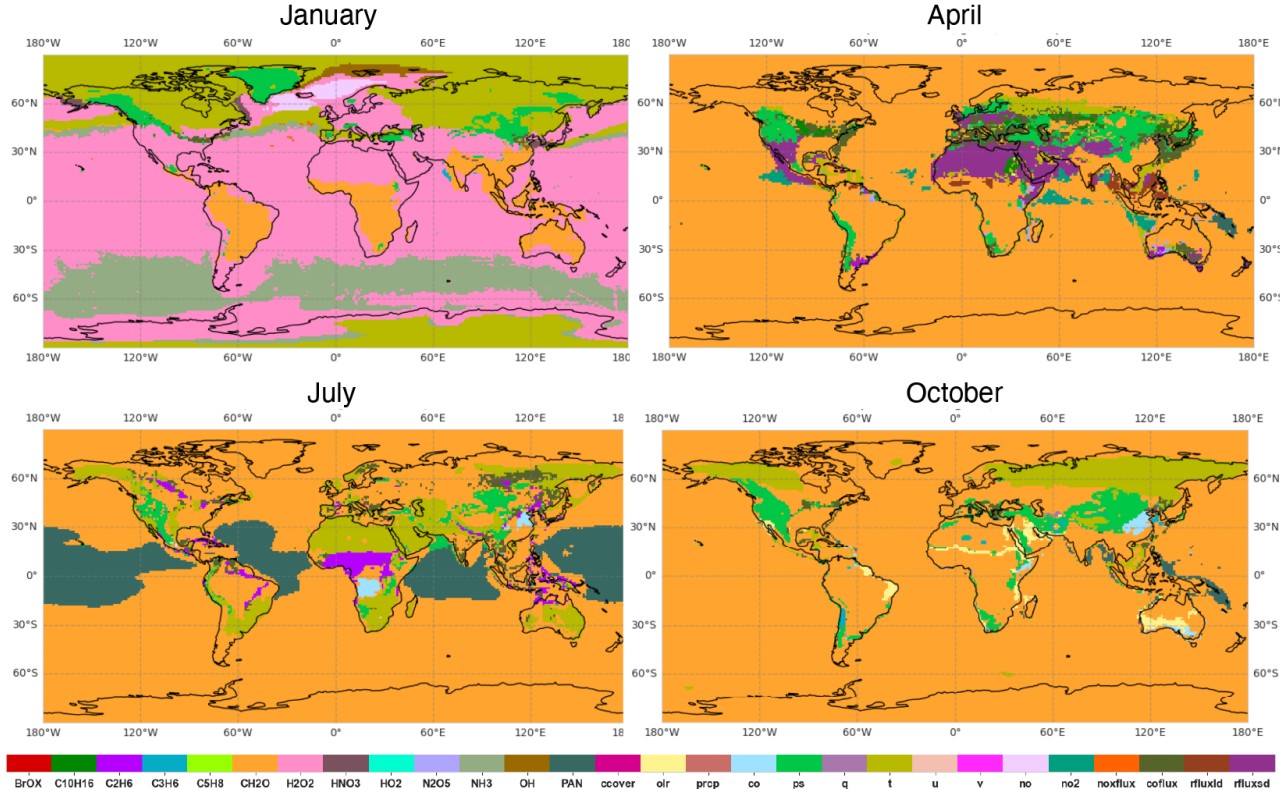

**Figure 10.** Spatial maps of the top contributors to the predicted ozone bias across all ML input variables for each location in January, April, July, and October.

the influence of transported precursors and photochemical processes. In areas with exceptionally high CO emissions, such as eastern China in October and central Africa in July, CO emerges as the dominant contributor, emphasizing the importance of accurately characterizing CO emissions and CO-related chemical processes in these areas. Similarly, $C_2H_6$ is identified as the dominant contributor over central Africa in July, which corresponds to intense biomass burning activities.

     While these findings on influential parameters provide valuable insights into the variability of ozone bias, their interactions

with other factors through complex chemical and physical processes present significant challenges for interpretation. Focusing solely on the most influential parameters may result in an oversimplification of the analysis, as these interactions often obscure essential underlying mechanisms. Furthermore, while ML-based attribution approaches provide detailed insights, they may exhibit abrupt temporal changes that are difficult to understand given our current scientific knowledge. The significance of these estimates is therefore questionable. These limitations underscore the need for further refinement of the ML methodology

to improve the reliability and interpretability of results.





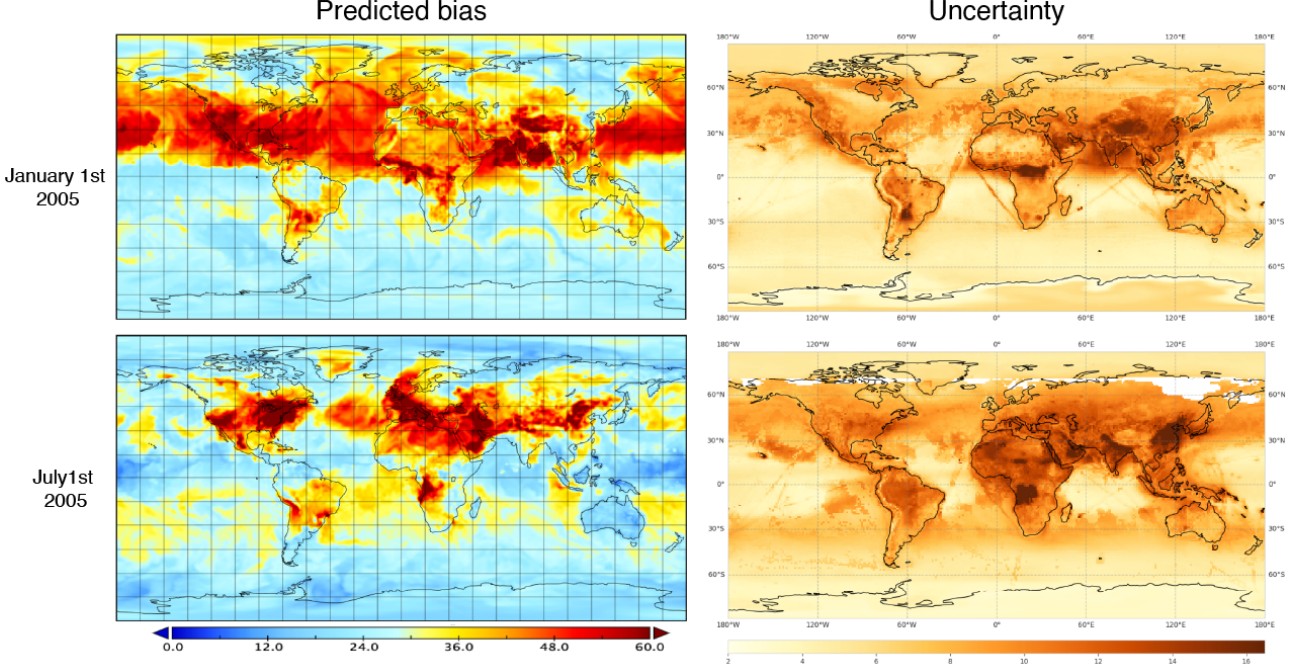

**Figure 11.** Spatial maps of ML-predicted ozone bias (left) and its associated uncertainty (right) for January 1st, 2005 (top panels), and July 1st, 2005 (bottom panels). The maps illustrate the regional variations in predicted bias and the corresponding confidence levels of the ML estimates.

## 5 Discussion

### 5.1 Uncertainty distributions

Uncertainty quantification (UQ) is essential for interpreting ML results. Incorporating comprehensive UQ into the ML framework provides direct insights into the confidence of the bias predicted. As illustrated in Fig. 11, the spatial and temporal patterns of estimated uncertainties are obvious, with larger uncertainties estimated over polluted regions. It it noteworthy that the spatial pattern of uncertainty exhibits some discrepancies from that of the predicted bias. For example, the relative uncertainty value in comparison to the predicted bias is lower over oceans but higher over land, particularly in the tropics and SH. These patterns align with potential error distributions identified in the emulator runs (Section 3.1). The uncertainty maps are of value in assessing the utility of bias-corrected ozone fields in informing ozone variations.

To further investigate uncertainty distributions, a local clustering analysis embedded within the ML framework was conducted using the mini-batch KMeans clustering algorithm (Sculley, 2011), which is a variant of the standard KMeans clustering algorithm (Lloyd, 1982) and uses mini-batches of data samples to improve computational efficiency while maintaining the same optimization objective. The mini-batch KMeans clustering is an iterative algorithm consisting of three major steps: (1) the random selection of data samples to form a mini-batch, (2) the assignment of each data sample to the nearest cluster cen-



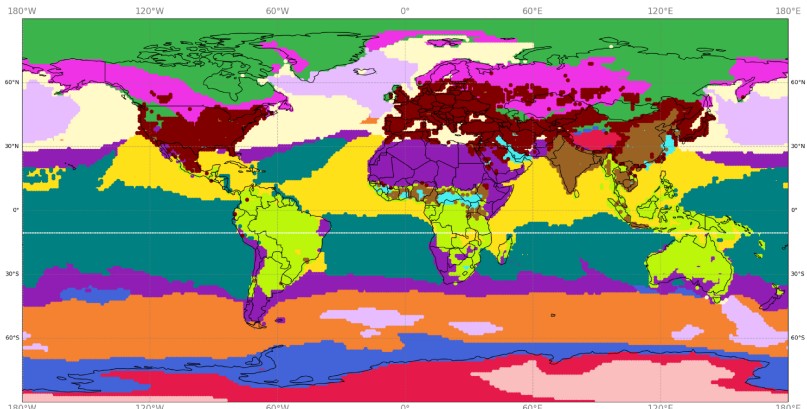

**Figure 12.** Local model clustering map of surface ozone on July 1st, 2005, estimated using MOMO-Chem chemical reanalysis outputs. The map illustrates spatially distinct regions grouped by similar ozone variability patterns and dominant contributing factors, with each color denoting a unique cluster.

troid with the least squared Euclidean distance, and (3) the updating of the cluster centroids for data samples assigned to each cluster. These steps are repeated until the assignments remain unchanged and the cluster centroids become stable, indicating convergence.

     The local clustering analysis categorized regions with similar ozone variability and driving factors. In the context of ML predictions, observational data are expected to impose similar constraints within each local area or among similar clusters, 505    leading to a common uncertainty distribution across grid points within the same cluster. The number of observations within a cluster is considered to be a critical determinant of ML prediction uncertainty. Regions with sparse or no observational data are likely to have less constrained ML predictions, resulting in higher associated uncertainties.

     As illustrated in Fig. 12, the cluster analysis revealed the existence of distinct regional ozone patterns, which appear to be influenced by a number of regional factors, including meteorological conditions, land use, population density, and industrial 510    activities. For example, the United States, Western Europe, and parts of East Asia were grouped into the same cluster, indicating that the ozone driving mechanisms are similar. The similarity between the regions also suggests that observational information from these regions can be shared in order to reduce the uncertainty of ML predictions within that cluster. The agreement between the spatial patterns of uncertainty distributions (Fig. 11) and the clustering analysis (Fig. 12) highlights the value of clustering in understanding the drivers of ML uncertainty. Furthermore, the clustering analysis can improve ML predictive 515    performance by identifying region-specific patterns and dominant factors, facilitating the development of localized models that better capture the unique dynamics of each region.





## 5.2 Challenges to scientific interpretation

The application of explainable ML at the process level is frequently constrained by the selection of input parameters, particularly when the input variable set is extensive. Silva and Keller (2024) emphasized the necessity for circumspection when applying explainable AI methods to datasets with highly correlated or dependent features. Such applications may yield spurious process-level explanations. They recommended that the current generation of explainable AI techniques be primarily used for understanding system-level behavior, with caution when applying them for process-level scientific discovery in physical sciences.

We encountered similar challenges. Some bias drivers identified by the explainable ML framework lacked scientific plausibility, particularly when a considerable number of input variables were included. This is likely attributable to the elevated probability of selecting spurious importance features among highly correlated variables. There are substantial correlations among chemically related species in the reanalysis outputs, with covariance patterns that vary substantially over time and space (figure not shown). Such correlations can introduce spurious signals into driver analyses, thereby complicating the interpretation of ML results. To address this issue, we conducted sensitivity analyses using ML to evaluate whether the input datasets avoided spurious signals while retaining the essential scientific information about bias drivers. Despite these efforts, ensuring robustness remains a significant challenge.

It is essential to validate the results of ML through the use of independent methodologies. For instance, CTM sensitivity experiments may be employed to introduce perturbations to the parameters identified as significant drivers by ML and then to evaluate their influence on ozone. For example, ML with a large number of input parameters identified $NH_3$ and methanol as significant contributors to ozone bias across diverse regions during specific months. Nevertheless, CTM simulations with a perturbation (e.g., by 10 %) in $NH_3$ or methanol showed only marginal impacts on ozone. Such discrepancies in their implications highlight the necessity for comprehensive validation prior to deriving to process-level insights from ML results.

Reducing the number of input variables, as conducted in this study through correlation analysis, and also with a focus on specific scientific objectives, can assist in minimizing these challenges. However, this approach may also restrict the potential to uncover unexpected scientific findings. Further advancements in explainable AI techniques are essential to fully leverage the comprehensive outputs from chemical reanalysis and CTMs, thereby enabling a more accurate and detailed understanding of bias drivers.

## 5.3 Different drivers of ozone and its model bias

Comprehensive analysis of factors influencing ozone variability can be conducted using CTM sensitivity experiments and source-receptor relationship analyses. These approaches provide detailed insights into the physical and chemical processes that drive ozone dynamics. However, these methods are computationally expensive and have limited capacity to assess the full range of potential drivers across different regions and timescales. In contrast, explainable ML offers a complementary perspective, providing instantaneous and comprehensive insights into the drivers of ozone variability across large datasets. Regarding model bias drivers, the information is limited due to the sparse distribution of validation data. ML can address this



limitation by providing detailed spatial and temporal information on both ozone concentrations and biases. Such insights are of great value in the improvement of physical models.

The primary drivers identified through ML demonstrate notable discrepancies in their impact on ozone concentrations and model bias. For example, BrOx was identified as a significant driver of surface ozone concentrations. However, its impact on ozone bias was found to be negligible (figure not shown). Similar inconsistencies were observed for other parameters, making it

challenging to fully comprehend the underlying reasons for these discrepancies. It is possible that poorly characterized model parameters, such as precursor emissions from biogenic or anthropogenic sources, may have a more pronounced impact on model biases than on variability. This indicates the necessity for further effort to provide their scientific interpretation of both drivers. It may also indicate the presence of spurious signals in the ML driver analysis, which also requires closer consideration and validation.

## 5.4   Implication for improving model, observation, and reanalysis

The current chemical reanalysis is constrained by limitations due to the reduced sensitivity of assimilated measurements toward the surface, which results in insufficient direct observational constraints on surface ozone. The assimilation of precursor species such as NOx and CO provides comprehensive constraints on the spatial and temporal patterns of surface ozone. However, certain reanalysis bias patterns were commonly found in CTM simulations that did not incorporate any DA. This indicates

that the bias driver information derived from chemical reanalysis can inform improvements in CTMs. Furthermore, these insights could be applied to correct biases in future ozone predictions (Liu et al., 2022). Nevertheless, ML does not provide guidance on how to modify model processes. Modifications to CTMs could entail the introduction of new chemical reactions, improvement or removal of outdated parametrization, or adjustment of parameters such as chemical reaction and photolysis rates. To ensure these updates are scientifically robust, proposed changes must align with existing knowledge derived from

laboratory experiments and observations not yet integrated into the model. Such ML-driven suggestions can direct targeted research efforts aimed at an improved understanding individual model processes, including new observational campaigns and detailed analyses of individual model components.

The bias driver analyses also point to additional observational constraints necessary for improving chemical reanalysis. Our previous studies have demonstrated that optimization of precursor emissions and assimilation of ozone and other species in the

upper troposphere and lower stratosphere has facilitated improvements in the ozone analysis for the entire troposphere, including near surface levels (Miyazaki et al., 2019). Nevertheless, the remaining bias highlights need to add observational constrains. Drivers such as $CH_2O$, identified as critical in various regions by ML, investigate the potential benefits of assimilating $CH_2O$ column measurements from instruments like OMI and TROPOMI to reduce reanalysis ozone bias. Similarly, the application of advanced tropospheric ozone retrievals with enhanced sensitivity to the lower troposphere (Fu et al., 2018; Okamoto et al.,

2023) could facilitate the improvement of the analysis of lower tropospheric ozone. Additionally, comprehensive outputs from DA, such as analysis ensemble spread, a measure of DA uncertainty, and analysis increment, a measure of adjustments by DA, can provide unique insights into the necessity for additional observational constraints. Integrating these DA statistics as inputs into ML frameworks could offer a potential avenue for more effectively identifying and addressing further improvements.



Sub-grid scale processes, such as urban-scale chemistry and planetary boundary layer (PBL) mixing (Ko et al., 2022), are likely significant contributors to model biases due to the coarse spatial resolution of the current reanalysis. Incorporating parameters related to sub-grid processes, such as vertical mixing rates, into the ML inputs could provide insight into their role as drivers of ozone bias. Moreover, preliminary ML tests confirmed that adding high-resolution satellite data, such as MODIS fire burnt areas and land use information, has the potential to improve the prediction of ozone bias, particularly during periods of extreme pollution (figure not shown). Further investigation is required to comprehend how the incorporation of high-resolution inputs enhance the ML performance and provides actionable insights for model improvement. Furthermore, the use of high-resolution models is crucial for reducing ozone biases (Skipper et al., 2024; Sekiya et al., 2021). ML-based down-scaling approaches could also be used to generate high-resolution fields from the coarse reanalysis outputs, offering a practical solution for applications such as health impact assessments.

## 6 Conclusions

Providing accurate global estimates of air pollution is crucial for evaluating the public health burden of diseases associated with air pollution exposure. This, in turn, informs effective environmental policy-making. However, current knowledge of air pollution is hindered by substantial biases in model predictions and limitations in the observational coverage of existing monitoring networks. While chemical reanalysis has significantly advanced our ability to reproduce regional and global ozone patterns, it remains fundamentally constrained by the model performance and the sparse spatial coverage of observations.

We utilized an explainable ML framework, based on a regression-tree randomized ensemble approach and TOAR observations, to analyze regional dependencies of ozone bias in the MOMO-Chem chemical reanalysis products. The results demonstrate that the developed ML framework effectively predicts ozone bias magnitude and spatial-temporal variations across diverse geographical regions, such as North America, Europe, and East Asia. Furthermore, it extends bias predictions to regions lacking surface observational networks, thereby providing a comprehensive global perspective on chemical reanalysis bias. By extracting and synthesizing local and global measures of how input parameters affect predicted bias, the ML framework facilitated model explanation and quantification of driver impacts. This approach yielded unique insights into the factors controlling biases in air quality assessments.

The analysis of ozone bias drivers revealed distinct spatial and temporal patterns, which highlighted the intricate interplay of meteorological conditions, chemical processes, and emissions. Surface pressure, temperature, and key chemical species such as $CH_2O$, PAN, $HNO_3$, and CO were identified as significant contributors, with their impacts varying across regions and seasons. $CH_2O$ was identified as a dominant factor in North America and East Asia, particularly during the summer months. This reflects its role in VOC-limited ozone regimes, which are driven by both anthropogenic and biogenic sources. In regions with complex topography, such as the Andes and the western United States, Surface pressure played a critical role, with its contribution varying seasonally. This indicates interactions with synoptic weather patterns and local dynamics. Notably, combustion-related emissions showed substantial contributions, particularly from CO and $C_2H_6$. The influence of CO emissions strongly on ozone bias was particularly evident in regions characterized by high industrial activity, such as eastern China, as well as in biomass



burning hotspots, including central Africa and Southeast Asia. Wildfires amplified ozone bias through CO, CH$_2$O, PAN, and VOCs, with notable impacts occurring over central Africa, South America, and Southeast Asia. Biogenic emissions, such as C$_5$H$_8$ also contributed significantly, particularly over forested regions like the Amazon, central Africa, and Southeast Asia. Additionally, radiation emerged as an important driver at low latitudes, reflecting its influence on photochemical reactions and atmospheric dynamics.

These findings highlight the diverse and region-specific contributions of meteorological conditions, combustion, wildfire, and biogenic sources to ozone bias. By pinpointing key contributors and their variability, this study provides a roadmap for targeted improvements in chemical transport models, DA systems, and emissions inventories, thereby facilitating a more precise representation of ozone patterns in chemical reanalysis. Such advancements are of critical importance for enhancing global air quality predictions and supporting informed pollution management policies. Conventional methods, such as sensitivity analyses using CTMs, require considerable computational resources to evaluate the contributions of each factor. In contrast, explainable ML offers a consistent and comprehensive alternative, capable of assessing the relative importance of diverse parameters across spatial and temporal dimensions. This adaptability allows the ML framework to be applied to other Earth system reanalyses and modeling, which can impact various areas of Earth Science. However, the complexity of interactions among various meteorological, chemical, and anthropogenic factors presents challenges in their interpretation and require rigorous validation of identified drivers against established scientific knowledge. By addressing these challenges, explainable ML will not only enhance our understanding of ozone bias, but also pave the way for actionable insights and lead to an improved framework for more effectively mitigating air pollution and its impacts.

*Code availability.* The ML code is publicly available at https://github.com/JPLMLIA/SUDSAQ

.

*Data availability.* The TROPESS chemical reanalysis product, TCR-2 data, as a part of the MOMO-Chem framework, was used in this study and is available at NASA GED DISC website at https://disc.gsfc.nasa.gov.

*Author contributions.* KM, KB, and YM designed the research; YM, JM, and SL conducted the ML calculations; KM provided the MOMO-Chem reanalysis data; KM, YM, JM, and SL analyzed the ML outputs; KM wrote the paper, with inputs from KB, YM, JM, and SL.

*Competing interests.* The authors declare that they have no conflict of interest.



*Acknowledgements.*  We acknowledge the use of data products from the National Aeronautics and Space Administration (NASA Aura and EOS Terra and Aqua satellite missions. We also acknowledge the support of the NASA Atmospheric Composition: Aura Science Team Program (19-AURAST19-0044), Earth Science U.S. Participating Investigator program (22-EUSPI22-0005), ACMAP (22-ACMAP22-0013), and the NASA TROPESS project. Part of this work was conducted at the Jet Propulsion Laboratory, California Institute of Technology, under contract with NASA.




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
