# Peer review of "Identifying Drivers of Surface Ozone Bias in Global Chemical Reanalysis with Explainable Machine Learning"

_EGUsphere, 2024_

## Author Response (AR1)

**Author's comments in reply to the anonymous referee for "Identifying Drivers of Surface Ozone Bias in Global Chemical Reanalysis with Explainable Machine Learning" by Miyazaki et al**

*Reply to Referee #1*

We would like to thank the referee for the helpful comments. We have revised the manuscript according to the comments, and hope that the revised version is now suitable for publication. Below are the referee comments in blue italics with our replies in normal font.

*General comments:*

1. *The method section in the current version of the manuscript can be improved. For example, it is sometimes confusing for readers to understand what the ML model is trying to predict. If the ground truth used to train the ML model is TOAR observations aggregated to the TCR-2 grid, then how is the global training and evaluation conducted?*

We appreciate the reviewer's comment regarding the methodology section, particularly concerning the ML prediction target and the global training and evaluation strategy. In the revised manuscript (Section 2.2), we have now clarified the following key points:

1. The ML model is trained using inputs from the MOMO-Chem chemical reanalysis and surface ozone observations from the TOAR network, which are spatially aggregated to the TCR-2 grid.
2. Once trained, the ML model is applied globally, including to grid cells without TOAR observations, allowing extrapolation of bias estimates beyond the observational network.
3. To assess ML prediction skill, we apply a leave-one-year-out cross-validation strategy. In each fold, one year of data is withheld across all grid cells, and the remaining 15 years are used for training. This maintains both temporal and spatial representativeness in evaluation.

The relevant sentences have been rewritten as follows:

*"To predict the reanalysis ozone bias, which is defined as the difference between the reanalysis and TOAR observations, using a given set of input variables, we employed a variant of the widely used ensemble tree method, Random Forest (RF)"*

*"The ML inputs included surface ozone data, MDA8, from the TOAR data sets, which served as the ground truth, along with outputs from the MOMO-Chem reanalysis. TOAR observations were aggregated to the MOMO-Chem reanalysis grid of 1.125° × 1.125° by computing the median value of surface ozone for all*

*stations within each grid box. This approach ensures a consistent spatial resolution between observations and reanalysis outputs. The reanalysis bias for each grid cell was then derived by comparing the MOMO-Chem reanalysis value with the corresponding median surface ozone value obtained from TOAR observations. Observations below 0 ppb or above 150 ppb were excluded to ensure data quality. For other reanalysis variables, daytime averages (8–15 local time) were derived from the 2-hourly reanalysis outputs and used in the ML calculations.*

*"To enhance computational tractability and avoid the influence of seasonality, particularly with regard to explainability metrics, we trained separate QRF models for each month. For training and evaluation, we employed a leave-one-year-out cross-validation strategy, where one year was withheld from the full dataset (2005–2020) across all grid cells, and the remaining 15 years were used for model training. This strategy ensured both temporal and spatial diversity was maintained in the training data."*

*"After training, the ML model was applied globally, including to grid boxes without TOAR observations, allowing us to estimate surface ozone biases and their drivers over the globe. This approach enabled global extrapolation of the learned bias patterns while maintaining a clear separation between training and evaluation domains."*

To further aid the reader's understanding of our methodology, we have added a new Figure 1 in the revised manuscript.

*"Figure 1. Schematic diagram of the machine learning (ML) framework used to predict surface ozone MDA8 bias in the MOMO-Chem chemical reanalysis. The framework integrates global reanalysis outputs with surface ozone observations from the TOAR network (top left). TOAR observations are spatially aggregated to the reanalysis grid to construct training data, and the reanalysis bias is calculated as the difference between the MOMO-Chem output and the aggregated TOAR observations. Separate Random Forest (RF) models are trained for each calendar month using a leave-one-year-out cross-validation approach over the 2005–2020 period (top right). The trained models are then applied globally to estimate surface ozone bias across all grid cells, including those without observational coverage (bottom right). Explainable ML techniques, including SHAP values, permutation importance, and spatiotemporal feature attribution, are then used to quantify prediction uncertainty and identify key drivers of the bias (bottom left)."*

2.  *It is mentioned around line 160 that the spatial resolution of 1.125 deg x 1.125 deg can limit the representativeness of gridded data sets. While I appreciate the discussion on the limitation, I am wondering if it is possible that the coarse resolution could be one of the uncertainty drivers? It would be great to see more details on this, for example, how many sites are in urban/rural regions and if there is any imbalance between urban/rural grid boxes. It is also not mentioned how urban and rural regions are defined*

We appreciate the reviewer's suggestion regarding the potential impact of spatial resolution and the urban–rural distribution of observational sites. According to the TOAR-I dataset (Fleming et al., 2018), the global network included 1,453 urban sites and 3,348 non-urban sites, as illustrated in the figure below.

[Figure]

Fig. Urban and non-urban sites based on warm season average ozone data for 2010–2014. As described in Fleming et al. (2018, https://doi.org/10.1525/elementa.273.f2), these classifications were derived using nighttime light intensity and population density thresholds.

These sites are often geographically contiguous, particularly around major metropolitan regions. As a result, it is common for both urban and non-urban sites to be located within the same 1.125° × 1.125° reanalysis grid cell. Both urban and non-urban sites are frequently located within the same 1.125° × 1.125° reanalysis grid cell. In core urban zones, the observational signal is typically dominated by urban sites, while in adjacent suburban and rural areas, non-urban sites may be more prevalent. This spatial mixing can obscure underlying differences in chemical regimes and emission characteristics, limiting the ML model's capacity to learn localized relationships when applied to coarsely aggregated data. While a detailed assessment of how the urban-to-non-urban site ratio within each grid cell influences ML prediction accuracy is beyond the scope of the present study, we acknowledge this as an

important avenue for future investigation. To acknowledge this issue explicitly in the manuscript, we have revised Section 3.2 to include the following statement:

*"In particular, spatial smoothing resulting from the relatively coarse resolution of the reanalysis can limit the ML model's ability to capture fine-scale chemical and dynamical processes, especially in urban environments. The aggregation of urban and non-urban chemical regimes within individual grid cells can introduce representativeness errors that add uncertainty to ML predictions. Depending on the magnitude and spatial variability of sub-grid processes, this may lead to systematic underestimation or overestimation of the reanalysis bias."*

   3. *A number of important figures mentioned in the discussion are not shown. It would become easier for readers to follow the discussion, if these figures are provided in supplementary materials and referenced in the main text.*

Thank you for the suggestion. To improve clarity and support the discussion, we have added the following figures to the Supplementary Materials:
   - Fig. S1a–d: Additional maps of ozone bias contributions from key variables.
   - Fig. S2: Correlation matrix of input variables over North America.
These figures are now referenced in the main text where relevant.

***Specific comments:***
*Line 80: (Watson et al., 2019)*

Fixed.

*Line 99-100: Incomplete sentence*

Added "is" before "to provide'

*Table 1: Are the meteorological variables from surface only and essentially identical to ERA-Interim data (i.e., they are not optimized in TCR-2, right?). Also, are all the chemical variables (concentrations and emissions) optimized in the data assimilation system, or only a number of the chemical variables are optimized?*

Yes, the meteorological variables listed in Table 1 (e.g., temperature, wind, pressure, radiation) are taken from the model's surface outputs and are not optimized in the TCR-2 data assimilation. The meteorological

fields are nudged toward ERA-Interim, and thus provide information nearly identical to that of ERA-Interim reanalysis. As for chemical variables, only a subset of species and emissions, such as $NO_2$, CO, and $SO_2$ are directly optimized through satellite data assimilation in TCR-2 (see Miyazaki et al., 2020). However, assimilation of these species exerts indirect constraints on many other chemical constituents through interactions in the chemical system. For instance, changes in OH induced by the multi-species assimilation can propagate to other reactive species, resulting in significant deviations from the model's free-running simulation. To clarify these points, we have added the following sentences in Section 2.1.1 of the revised manuscript:

*"As described above, the meteorological variables used are obtained by nudging the model's meteorological fields toward ERA-Interim reanalysis data. A subset of chemical species and emissions (e.g., $NO_2$, CO, $SO_2$) are directly constrained by satellite observations. Observational information also propagates through model chemical processes (e.g., via OH perturbations), which enables indirect optimization of other species, leading to reanalysis fields that can differ significantly from those of model simulations without data assimilation."*

*Line 170: Does this mean you would need to provide both the mean and quantile values from the ground truth while training the QRF?*

No, it is not necessary to provide both the mean and quantile values during training. The QRF algorithm requires only the original target variable as input. Rather, the quantiles are computed by QRF from the distribution of predictions across the ensemble of decision trees. To clarify this point, the relevant sentences in Section 2.2.1 have been rewritten as follows:

*"Specifically, we implemented Quantile Random Forest (QRF), which modifies the loss function to predict both the mean and quantile values of the conditional distribution. The quantile outputs provided by QRF can be used to estimate prediction uncertainties."*

*Line 190: I thought one of the advantages of Permutation Importance is that you can calculate PI using already trained models and avoiding any re-training?*

Thank you for the comment. You are correct. One of the key advantages of Permutation Importance (PI) is that it can be computed without retraining the model. Instead, it evaluates the impact of each input variable on the model's predictive performance by perturbing input features in an already trained model. To clarify this, we have revised the description in Section 2.2.2 as follows:

*"PI is a model-agnostic metric that evaluates the contribution of individual input variables by randomly permuting one variable at a time. The trained model then makes predictions on the permuted data, and the resulting change in predictive accuracy, typically assessed using metrics such as root mean squared error (RMSE), is computed. This method does not require any re-training of the model, making it both computationally efficient and suitable for interpreting complex models. A larger drop in accuracy indicates greater importance of the permuted variable, independent of the effects of other inputs. While PI does not account for cross-correlations between input variables, it can identify independent relationships and highlight inter-variable dependencies."*

*Section 2.3.1: The purposes of the two emulator runs are still not very clearly written in the current version. Are the emulator runs referring to first training using global TCR-2 data and then training using regional TOAR data? Also for the Emu_toar run, do you still use global MDA8 in the evaluation, because the goal is to test the generalizability of the model?*

In the Emu_gl experiment, the emulator is trained using global TCR-2 reanalysis inputs (excluding MDA8 itself as a predictor) across all available grid cells. In the Emu_toar experiment, the training is restricted to only the regions covered densely by TOAR observations (North America, Europe, and East Asia), to simulate the realistic constraint of observational coverage. In both experiments, the evaluation is conducted globally, using the true global MDA8 reanalysis field as the target for testing. The primary goal of Emu_toar is thus to assess the generalizability of a model trained only on observationally dense regions when applied globally. To address this more clearly, we have rewritten the relevant sentences in Section 2.3.1 as follows:

*"The ML framework was first evaluated in emulation mode to reproduce the reanalysis MDA8 fields. By leveraging the true global MDA8 fields provided by the reanalysis for evaluation, this framework allowed for an assessment and optimization of the baseline ML performance. Two emulator runs were conducted.*

*The first experiment (Emu_gl) trained the ML model using global reanalysis fields (excluding MDA8 itself from the input features) to emulate ozone distributions under full data coverage. This configuration demonstrates the ideal predictive performance of the ML framework when comprehensive information is available.*

*The second experiment (Emu_toar) restricted the training data to North America, Europe, and East Asia, where TOAR observational coverage is dense. This configuration enables the assessment of the impact of limited observational coverage on the model's ability to represent global ozone distributions.*

*The TOAR sampled area encompassed North America (20°N to 55°N, 125°W to 70°W), Europe (35°N to 65°N, 10°W to 25°E), and East Asia (20°N to 50°N, 100°E to 145°E).*

*In both experiments, the evaluation was conducted globally against the true reanalysis MDA8 fields, allowing for a consistent assessment of the model's generalization capability under both dense and sparse observational coverage scenarios."*

The sentence has been rewritten as

*"However, notable discrepancies were found in the central Pacific, where relative errors exceeded 30% and absolute errors were greater than 12 ppb. These discrepancies over the limited regions indicate the presence of local ozone-driving mechanisms that are insufficiently captured by the global statistics."*

*Figures 1 and 3: The caption says blue and red lines represent observed (actual) and ML-predicted values. But the legends indicate the opposite. My guess is orange lines are actual (i.e., legends are correct)?*

Corrected the captions. Thank you.

*Figure 4: Are the results from the independent out-of-training samples? For the bottom row, the full time series of actual and ML-predicted surface ozone biases seem to match pretty well. However, why are the correlation coefficients so low? Also, why do the North American results contain two predicted biases with zero values?*

Thank you for the detailed comments. The results shown are based on the leave-one-year-out cross-validation strategy, as more clearly described in the revised manuscript. We have updated the figure to exclude data gaps that previously appeared as zero values in the North American results. These zero entries were not valid predictions but instead corresponded to missing data and have now been removed from the plot. Also, in the original figure, the reported R values did not represent temporal correlation coefficients, which may have caused confusion. To address this, we have removed the R values from the revised figure. Instead, we now provide a more accurate description of the model performance in the revised manuscript text as follows:

*"The ML predictions well capture the temporal patterns of the actual bias at the regional scale, with temporal correlations of 0.98 for North America, 0.89 for Europe, and 0.85 for East Asia."*

*Figure 8: Do negative contributions to ozone bias mean that over some regions the corresponding parameters help reduce bias, or that these parameters lead to negative bias?*

Yes, negative contributions indicate that the variable tends to reduce ozone overprediction (in cases of positive bias) or amplify underprediction (in cases of negative bias). Conversely, positive contributions implies that the variable is associated with an increase in the predicted positive bias. To clarify this interpretation, we have revised both the figure caption and the relevant description in Section 4.2 as follows:

*"Negative contributions indicate that the variable tends to reduce overpredicted ozone bias (when the bias is positive) or amplify underpredicted ozone bias (when the bias is negative). Conversely, positive contributions suggest that the variable is associated with an increase in the positive bias or a reduction in the magnitude of the negative bias."*

*Line 468: is also a critical factor*

Corrected.

*Line 490: It is noteworthy*

Corrected.

*Figure 12: What are the dominant contributing factors for each color?*

Thank you for the comment. Identifying the dominant contributing factors in this figure could indeed support the interpretation of local model behavior. However, the spatial distribution of dominant contributors has already been thoroughly analyzed and visualized in Figure 11 (in the revised manuscript). Figure 13 (in the revised manuscript) is specifically designed to highlight the spatial clustering of surface ozone characteristics. Including attribution in this figure would introduce redundancy and potentially obscure its intended focus. To maintain clarity and ensure the figures serve complementary purposes, we have chosen not to include dominant contributor information in Figure 13.

**Author's comments in reply to the anonymous referee for "Identifying Drivers of Surface Ozone Bias in Global Chemical Reanalysis with Explainable Machine Learning" by Miyazaki et al**

*Reply to Referee #2*

*The authors use an RF algorithm to (1) emulate predicted concentrations of surface ozone from a leading tropospheric chemical reanalysis product, and (2) predict the bias of the reanalysis product relative to global surface observations. The authors use explainable AI techniques to understand drivers of bias in ozone reanalysis. These results offer a useful perspective on O₃ chemical transport model predictions and data assimilation output, and I recommend publication after the following comments are addressed. Major points:*

- *My biggest concern by far in this work is spatial extrapolation: the TOAR surface data used in training are clustered in a few regions (North America, Europe, and east Asia) while bias is predicted globally. The authors are aware of this, but spatial crossvalidation is a more direct way of quantifying the issue and is not done in this work. I am most concerned about (1) oceans, (2) boreal regions, and (3) the tropics where training data is limited. Consider withholding some of the few training sites in these regions and measure how well the RF predicts bias there (the clustering maps in Figure 12 might be a reasonable way to do crossvalidation). The authors could also use more recent observations in China and India for independent evaluation. Do we really have enough data to use RF, a highly data-dependent algorithm, for extrapolation to these regions?*

We thank the reviewer for raising this important concern regarding spatial extrapolation. We fully agree that evaluating the model's ability to generalize beyond observationally dense regions is critical for assessing the robustness of our approach. In response, we have taken the following steps and provided clarifications:

**1. Assessment through emulator runs:**

To examine the impact of sparse observational coverage, we performed emulator experiments in which the ML model was trained only on TOAR-covered regions and evaluated globally. These results revealed that:

- Prediction errors were systematically higher in the tropics, oceans, and high-latitude boreal regions compared to the NH midlatitude regions with dense TOAR coverage.
- Oceanic regions exhibited particularly large uncertainties, consistent with the absence of surface constraint.

- These outcomes highlight the degradation of predictive skill in observationally sparse regions and demonstrate the intrinsic limitations of spatial extrapolation in such settings, even in the absence of formal spatial cross-validation.

2. **Justification of temporal cross-validation:**

In the ML runs using actual observations, we employed a leave-one-year-out **temporal cross-validation** strategy rather than spatial cross-validation. This decision was based on the sparsity and irregular distribution of TOAR sites, which made it difficult to define spatially independent and statistically robust validation subsets. Spatial cross-validation under such conditions would risk creating training or validation sets that are too small or not representative of broader spatial patterns. While this limits our ability to assess true spatial extrapolation, temporal cross-validation allows us to evaluate the model's generalizability across time, which is still relevant for long-term bias analysis.

We now explicitly discuss this in the revised manuscript in Section 2.3, where the cross-validation strategy is described as follows:

*"While the temporal cross-validation approach, implemented through a leave-one-year-out strategy, does not fully address the challenge of spatial extrapolation, it provides a robust framework for evaluating the model's generalization across years with diverse chemical and meteorological conditions. We acknowledge that spatial cross-validation would offer a more direct assessment of the model's extrapolation capability. However, this was not feasible in our case due to the sparse and uneven distribution of TOAR monitoring sites, particularly outside of North America, Europe, and East Asia, which results in limited spatial coverage and strong regional clustering. In many under-sampled regions, such as the tropics, boreal zones, and the Southern Hemisphere, the lack of contiguous observational clusters prevents the construction of spatially independent and statistically meaningful training and validation sets. Consequently, we relied on temporal cross-validation to preserve both data representativeness and model stability, while recognizing that spatial extrapolation remains an important area for future investigation. To complement this limitation, the ML's predictive performance in observationally sparse regions is further evaluated through dedicated emulator experiments described in the following section."*

3. **Limitations of Random Forest for extrapolation:**

We also acknowledge the fundamental concern raised by the reviewer regarding the suitability of RF for extrapolation. As an ensemble method, RF tends to interpolate within the convex hull of the training data and is not inherently designed to predict beyond regions with observational coverage. To clarify this limitation, we have added the following statement to Section 5.1:

*"In addition to the data imbalance, RF itself has inherent limitations in extrapolation. As an ensemble tree-based method, RF primarily interpolates within the convex hull of the training data and lacks the ability to generalize to regions with little or no observational coverage. Consequently, predictions over sparsely observed areas, such as the tropics and oceans, are subject to greater uncertainty and should be interpreted with caution. These algorithmic and data-related constraints underscore the need to expand global monitoring networks and explore hybrid approaches that integrate physical knowledge with ML."*

4. **On the use of recent observations in China and India:**

We appreciate the reviewer's suggestion to incorporate more recent observations from China and India for independent evaluation. At the time of this study, these data had not yet been fully integrated into the TOAR database. We agree that including such data would enhance the spatial representativeness of the training set and reduce extrapolation bias. We have addressed this point in the revised manuscript with the following statement in Section 5.1:

*"In addition, surface ozone observations from emerging monitoring networks, including those in China and India, were not yet fully incorporated into the TOAR database at the time of this study. Their inclusion in future work is expected to improve spatial representativeness, reduce extrapolation bias, and strengthen the reliability of ML-based inference in currently under-sampled regions."*

- *In cases of highly imbalanced training sets, where some regions are far overrepresented, methods like SMOTE or weighted training are sometimes employed to ensure that the RF is penalized more heavily for bad predictions at some sites. Did the authors consider using such approaches?*

Thank you for this valuable comment. We acknowledge that the spatial distribution of TOAR sites is highly imbalanced, which may introduce regional biases and limit the model's ability to generalize to globally diverse conditions. While SMOTE is designed primarily for classification problems, techniques such as weighted training or stratified sampling can be effective in addressing imbalance in a regression setting. In the present study, we prioritized model interpretability and transparency, and therefore did not implement these strategies. However, we agree that this is an important direction for future work,

particularly as more globally representative observational data become available. To reflect this point, we have added the following sentences to the discussion section (Section 5.1), highlighting the potential role of weighting methods:

*"We also note that the spatial distribution of training data is highly imbalanced. This imbalance may lead to an overrepresentation of region-specific patterns in the learned relationships, potentially limiting model generalizability. Although we did not implement weighting or rebalancing strategies such as region-based sampling weights or stratified training in this study, such techniques may offer an effective means of mitigating spatial biases in future applications."*

- *Explainable AI methods are vulnerable to collinearity in the inputs, as the authors are aware, and of the algorithms used SHAP (TreeExplainer) is most robust to this problem. I would like to see more comparison between SHAP and the other methods. For example, in Figure 6, what does SHAP suggest are the top contributors to ozone bias in these regions? In the literature, for SHAP regional attribution some authors use separate RFs trained on training sets focused on particular regions.*

[Figure]

Fig. Comparison of feature attribution metrics derived from SHAP and Conditional Feature Contributions (CFC). Bars represent normalized contributions of the variables to the surface ozone bias.

Thank you for this insightful comment. As shown in the comparison figure, the feature attribution metrics derived from SHAP and Conditional Feature Contributions (CFC) are highly consistent across regions. This agreement is expected for tree-based models such as RF, where both methods yield additive explanations by decomposing predictions into contributions from individual input variables. While SHAP uses a game-theoretic framework that averages over all possible feature combinations, CFC calculates importance based on the actual traversal paths of each instance through the decision trees. In settings with moderate collinearity and limited feature interactions, the two approaches tend to produce similar results

(Lundberg et al., 2020). The strong alignment observed in our study supports the robustness of the derived feature importance rankings. We have added a brief discussion of this comparison in Section 4.1 of the revised manuscript.

"We also compared the feature attribution results with SHAP values (not shown). In particular, SHAP and CFC yielded closely aligned rankings of the dominant contributors to surface ozone bias across regions. This agreement is expected for tree-based models, as both SHAP's game-theoretic averaging and CFC's path-based decomposition provide additive explanations. The consistency between these two approaches, especially under conditions of moderate input collinearity (Lundberg et al., 2020), supports the robustness of our feature importance analysis."

*Minor points:*
- *Figure 3: It is not surprising that RF has trouble predicting the distributional tails; this has long been observed in the literature (and is to be expected given it is an ensemble algorithm). Consider commenting on the limitation of this method for e.g. improving models such that they give better predictions of NAAQS ozone exceedances (e.g. MDA8).*

We agree that RF models are known to underperform in capturing the extremes of the target distribution, largely due to the averaging nature of ensemble predictions, which is a limitation well documented in the literature. This tendency is also evident in our results. To clarify this point, we have added a short discussion in Section 5.2 along with citations to relevant studies (Gao et al., 2022; Chen et al., 2021) that report similar findings.

"The ML predictions systematically underestimate the variability in surface ozone bias across all regions, indicating an underestimation of the occurrence of extreme (both positive and negative) bias values. This behavior is a well-known limitation of RF, which tend to underpredict distributional tails due to their ensemble averaging structure (Gao et al., 2022; Chen et al., 2021). Such underestimation is particularly relevant when aiming to detect exceedances of air quality standards, where accurate representation of high-ozone events is critical."

- *Given the given tropical Pacific pattern in RMSE (Figure 2) I am curious about the role of ENSO in driving RF error. Is lightning NOx a problem here?*

We have not explicitly evaluated interannual variability or the influence of ENSO in this study, so its role in driving RF prediction errors remains unclear. In addition to lightning $NO_x$, ENSO-related changes in

convection, cloud cover, and large-scale atmospheric circulation could indirectly affect surface ozone and contribute to the observed regional error patterns. Investigating these mechanisms, particularly through targeted analysis of interannual variability, represents a valuable direction for future research.

- *Could you clarify if TOAR surface sites averaged to the grid of the TCR-2 output? In places with many monitors within a single grid cell this could lead to sample bias where e.g. urban areas are even more disproportionately represented.*

Yes, TOAR surface ozone observations were aggregated to the 1.125° × 1.125° grid used in the TCR-2 reanalysis. In grid cells containing multiple monitoring sites, we used the median of all available TOAR observations to represent the surface ozone value, thereby reducing sensitivity to outliers and localized effects, particularly in urban environments. In regions with dense urban monitoring networks, this aggregation may introduce sample bias by over-representing urban conditions relative to the true grid-scale chemical environment. While this limitation cannot be entirely eliminated given the current observational distribution, we now address this issue explicitly in the revised manuscript. The following statement has been added to Section 3.2:

*"In particular, spatial smoothing resulting from the relatively coarse resolution of the reanalysis can limit the ML model's ability to capture fine-scale chemical and dynamical processes, especially in urban environments. The aggregation of urban and non-urban chemical regimes within individual grid cells can introduce representativeness errors that add uncertainty to ML predictions. Depending on the magnitude and spatial variability of sub-grid processes, this may lead to systematic underestimation or overestimation of the reanalysis bias."*

- *Figure 5: consider using same colorbar for observations and for predictions.*

Applied

- *Figure 11: consider also showing uncertainty as percentage of predicted bias*

Thank you for the suggestion. We considered showing uncertainty as a percentage of the predicted bias. However, in regions where the predicted bias is small, the relative uncertainty becomes disproportionately large and may misrepresent the true model behavior. For this reason, we chose to present the uncertainty in absolute terms, which more consistently reflects the magnitude of prediction spread across all regions.

- *Throughout, increase font size of figures. It can be quite hard to read.*

We have increased the font size in some figures to improve readability.

- *Some typos throughout. Here are a couple: Line 83: "the simulation of simulate" should read "the simulation of". Line 241: Missing unit after "exceeded 30" (I think it should be percent)*

We have carefully re-checked the manuscript and corrected some typos.